

# Retention of α-pinene oxidation products and nitro-aromatic compounds during riming

Christine Borchers[1], Jackson Seymore[2], Martanda Gautam[2], Konstantin Dörholt[3], Yannik Müller[1], Andreas Arndt[2], Laura Gömmer[2], Florian Ungeheuer[3], Miklós Szakáll[2], Stephan Borrmann[2,4], Alexander Theis[4], Alexander L. Vogel[3], Thorsten Hoffmann[1]

[1]Departmet of Chemistry, Johannes Gutenberg University, Mainz, Germany
[2]Institute for Atmospheric Physics, Johannes Gutenberg University, Mainz, Germany
[3]Institute for Atmospheric and Environmental Sciences, Goethe-University Frankfurt/Main, Germany
[4]Particle Chemistry Department, Max Planck Institute for Chemistry, Mainz, Germany

*Correspondence to*: Thorsten Hoffmann (t.hoffmann@uni-mainz.de)

**Abstract**

Riming is an important growth process of graupel and hailstones in mixed-phase zones of clouds, during which supercooled liquid droplets freeze on the surface of ice particles by contact. Compounds dissolved in the supercooled cloud droplets can remain in the ice or be released to the gas phase during freezing, which might play an important role in the vertical redistribution of these compounds in the atmosphere by convective cloud processes. This is important for estimating the availability of these compounds in the upper troposphere, where organic matter can promote new particle formation and growth. The amount of organics remaining in the ice phase can be described by the retention coefficient. Experiments were performed in the Mainz vertical wind tunnel under dry and wet growth conditions (temperature from $-12$ to $-3$ °C and a liquid water contents (LWC) of $0.9 \pm 0.2$ g m$^{-3}$ and $2.2 \pm 0.2$ g m$^{-3}$) as well as different pH values (4 and 5.6) to obtain the retention coefficients of α-pinene oxidation products and nitro-aromatic compounds. For cis-pinic acid, cis-pinonic acid, and (−)-pinanediol mean retention coefficients of $0.96 \pm 0.07$, $0.92 \pm 0.11$, and $0.98 \pm 0.08$ were obtained. 4-Nitrophenol, 4-nitrocatechol, 2-nitrobenzoic acid, and 2-nitrophenol showed mean retention coefficients of $1.01 \pm 0.07$, $1.01 \pm 0.14$, $0.99 \pm 0.04$, and $0.16 \pm 0.10$. Only the retention coefficient of 2-nitrophenol showed a dependence on temperature, growth regime, and pH. This is in accordance with previous studies which showed a dependence between the dimensionless effective Henry's law constant $H^*$ and the retention coefficient for inorganic and small organic molecules. Our results reveal that this correlation can also be applied to more complex organic molecules, and that retention under these conditions is negligible for molecules with $H^*$ below $10^3$, while unity retention can be expected for compounds with $H^*$ above $10^8$.

## 1 Introduction

The observation of a large number of small particles at high altitudes, which has already been made several times, is attributed to the formation of new particles (New Particle Formation, NPF) through the process of homogeneous nucleation and early



growth (Xiao et al., 2023; Williamson et al., 2019; Andreae et al., 2018; Kerminen et al., 2018). A common explanation for the presence of these particles is the uplift of condensable vapors with simultaneous removal of existing aerosol particles in deep convective clouds. This removal reduces the sinks for small particles and condensable vapors, supporting NPF (Clarke et al., 1998). However, Williamson et al. (2019) also showed that tropical convection does not produce these sinks. They then

argue that most of the models used underestimated available organic matter at high altitudes and predict less NPF in these regions. It is therefore important to investigate the possible transport mechanism of organic precursor components that could lead to NPF at high altitudes (Bardakov et al., 2022). Figure 1 shows vertical transport mechanisms of intermediate-volatile and semi-volatile organic compounds (IVOC and SVOC).

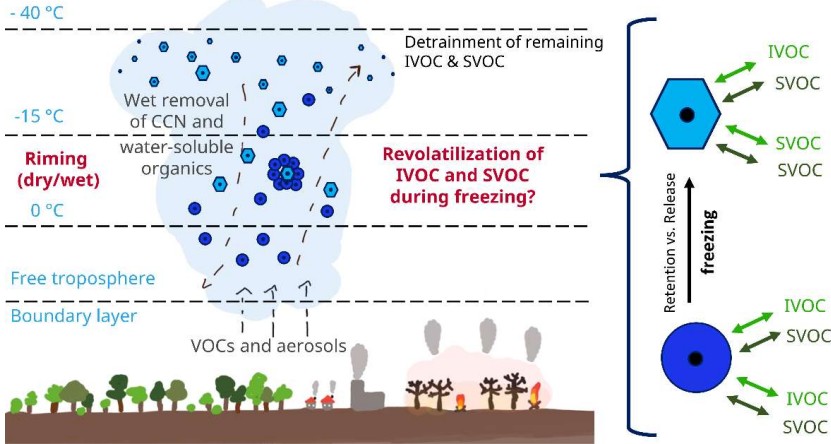

**Figure 1: Water-soluble intermediate-volatile and semi-volatile organic compounds (IVOC and SVOC) can be dissolved in water droplets (dark blue circles), which can be transported upwards by deep convection. In the mixed-phase zone of the clouds, the water droplets can collide with ice particles (light blue hexagon), resulting in riming.**

Organic compounds in the atmosphere can be categorized into different groups depending on their source, such as biogenic, anthropogenic, or compounds originating from the combustion of biomass (Gouw and Jimenez, 2009). Nitro-aromatic

compounds can be formed directly from the combustion of coal or wood, but can also be formed as secondary products from the reaction of phenols or cresols with NOx (Wang et al., 2020; Harrison et al., 2005). Terrestrial vegetation emits large quantities of volatile organic compounds (VOCs) such as isoprene and various monoterpenes (MT), with the most important MT α-pinene contributing around one third of global MT emissions (Sindelarova et al., 2014). In the atmosphere, oxidation by OH radicals, ozone or $NO_3$ radicals results in various products spanning orders of magnitude in volatility. Products such as

2-methyl tetrols, pinanediol, terpenylic acid, pinonic acid, pinic acid, have been described as major oxidation products (Kołodziejczyk et al., 2020; Bianchi et al., 2019; Nozière et al., 2015; Müller et al., 2012; Kroll and Seinfeld, 2008; Claeys et al., 2004; Hoffmann et al., 1997). At standard conditions, the saturation vapour pressure of these oxidation products is too



large for being relevant for new particle formation. Highly oxygenated organic molecules (HOMs) exhibit a sufficient low vapour pressure for NPF (Bianchi et al., 2019), however, their formation via autoxidation is suppressed at low temperatures

(Stolzenburg et al., 2018). Hence, the higher-volatility major oxidation products of isoprene and monoterpenes might be relevant for NPF in the upper troposphere, where HOM formation is suppressed and the colder temperatures cause saturation of the major oxidation products, which are classified as SVOCs at standard conditions. Both classes of compounds focused on in this study, i.e. pinene oxidation products and nitro-aromatic compounds, are IVOCs (saturation mass concentration $C^*$: $300 < C^* < 3 \cdot 10^6$ µg m$^{-3}$) or SVOCs ($0.3 < C^* < 300$ µg m$^{-3}$) (Simon et al., 2020; Andreae et al., 2018).

Water-soluble IVOCs and SVOCs can be dissolved in water droplets (dark blue circles in Figure 1), which can be transported upwards via deep convection. In the mixed-phase zone of clouds, retention or release of IVOCs and SVOCs during riming can occur. Riming describes the collision and freezing of supercooled water droplets on the surface of hydrometeors such as a graupel or snowflakes (light blue hexagon in Figure 1) which leads to their growth. The organic compounds can be trapped in the ice phase and then washed out by precipitation, or they can return to the gas phase by volatilization during freezing. This

revolatilization leads to a vertical redistribution in the atmosphere and could explain the occurrence of semi-volatile organic compounds at high altitudes in regions with deep convection. However, if the organic substances remain in the ice phase during freezing, they could also be transported further upwards and released into the gas phase by sublimation of the ice particles there (Pruppacher and Klett, 2010; Snider and Huang, 1998).

The proportion of the compound that remains in the ice can be described by the retention coefficient $R$, which indicates the

relative fraction of the trapped compound with a value between 0 and 1 (Bela et al., 2018; Stuart and Jacobson, 2003; Snider et al., 1992; Iribarne and Pyshnov, 1990). The retention of a compound is influenced by its chemical properties, such as the dimensionless effective Henry's law solubility constant $H^*$, as well as physical parameters such as temperature, droplet size, liquid water content in the cloud, ventilation, and potentially the pH of the droplet. Compounds with a small $H^*$ are more likely to return to the gas phase during riming, which results in a lower retention coefficient. In addition, external conditions have a

greater influence on retention for these kind of compounds, which is in contrast to compounds with high $H^*$ (Cuchiara et al., 2023; Jost et al., 2017; Stuart and Jacobson, 2004, 2003).

Previous measurements of inorganic and small organic species in the wind tunnel in Mainz confirm the correlation between $H^*$ and $R$ as well as the stronger dependence of retention on external factors such as temperature for lower $H^*$. Hydrochloric acid, nitric acid, malonic acid, and oxalic acid are characterized by a high $H^*$ value and remain completely in the ice phase

during freezing. Compounds with more moderate $H^*$ values such as ammonia, hydrogen peroxide, formic acid, and acetic acid have retention factors of $0.92 \pm 0.21$, $0.64 \pm 0.11$, $0.68 \pm 0.09$, and $0.72 \pm 0.16$. The organic compounds show a dependence on temperature and ventilation. Additionally, sulfur dioxide shows both the lowest $H^*$ value and retention coefficient ($0.46 \pm 0.16$) of the compounds discussed as well as with a dependence on external conditions (Jost et al., 2017; v. Blohn et al., 2013; 2011).

Up to now, only inorganic, and small organic molecules have been investigated with regard to their retention during the freezing process. Measurements of rain and cloud water have already shown that they contain α-pinene oxidation products



and nitrophenols (Spolnik et al., 2020; Desyaterik et al., 2013; Ganranoo et al., 2010). It is therefore obvious that these compounds are also present in the supercooled droplets within mixed phase zones of clouds. In contrast to the smaller organic molecules and inorganic compounds, the retention coefficients for α-pinene oxidation products and nitrophenols are unknown.

In this study, therefore, the retention coefficients of three α-pinene oxidation products (pinonic acid, pinic acid, and pinanediol) and four nitro-aromatic compounds (2-nitrobenzoic acid, 4-nitrocatechol ,4-nitrophenol, and 2-nitrophenol) are investigated in a series of wind tunnel experiments under simulated atmospheric conditions.

## 2 Experimental procedures

### 2.1 Mainz vertical wind tunnel

The experiments were carried out in the vertical wind tunnel of the Johannes Gutenberg University of Mainz, shown schematically in Figure 2. Here, hydrometeors ranging from a few tens of micrometers to centimeters can float freely in a vertical air stream at their terminal fall velocity. The prevailing conditions such as ventilation, mass, and heat transfer are very similar to those in the atmosphere (Pruppacher and Klett, 2010). A vacuum pump continuously sucked dried ambient air through the system to produce an air flow in the wind tunnel. For riming experiments the tunnel was cooled down to −18 °C

and water droplets were produced using up to four spray nozzles. These droplets were transported via the airflow to the experimental region (red circle, zoomed in: red rectangle in Figure 2). In the experimental region the supercooled droplets collided with three different surfaces—a simulated graupel, a liquid nitrogen finger, and Teflon-coated bars— and froze on the substrate. A detailed description of the experiments is provided in Sect. 2.3 Retention measurements. Further details on the wind tunnel can be found in two reviews by Diehl et al. (2011) and Szakáll et al. (2010).



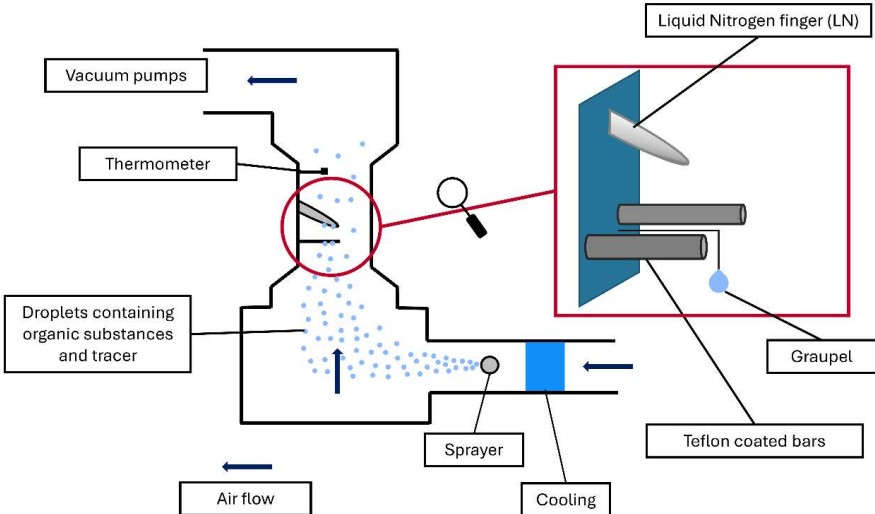

**Figure 2: Schematic of the wind tunnel. Cooled air transported the generated water droplets containing the compounds into the experimental region (red circle). Red rectangle: The enlarged experimental area shows the three surfaces on which the riming took place: Graupel, liquid nitrogen finger and Teflon-coated bars.**

### 2.2 Growth regimes

For riming, a distinction is made between dry and wet growth conditions, which are determined by the combination of temperature, liquid water content and ventilation. During freezing latent heat is released, which warms the surface of the rime collectors. Under dry growth conditions the surface temperature of the rime collector remains well below 0 °C and all the accreted cloud water freezes instantaneously on the rime collector, preserving a close to spherical shape. The surface temperature increases with increasing liquid water content, droplet size, and collision frequency of the droplets with the hydrometeor. If the temperature rises to a maximum value of 0 °C wet growth conditions are reached. At this point, not all the water that has collided with the rime collector freezes immediately. Typically, a liquid layer is formed which freezes very slowly and thus creates a dense ice structure (Pruppacher and Klett, 2010; Macklin, 1961; List, 1960).

The earlier wind tunnel measurements on retention coefficients (Jost et al., 2017; v. Blohn et al., 2013; v. Blohn et al., 2011), focused solely on retention during dry growth conditions. Thus, there remained the question to which extent wet growth conditions affect retention. For example, Michael and Stuart (2009) found in their theoretical study that during wet growth conditions $H^*$ is not a dominant factor and low retention values were also found for compounds like HCl. Here other factors like the supercooling of the liquid surface water are major determinators for the extent of retention. In the present study, the measurements were carried out at dry and wet growth conditions. However, unlike the study of Michael and Stuart (2009) the droplets did not shed off during these experiments. Calculations of the surface temperature during the growth of graupel reveals



that under our wet growth conditions the surface temperature varied between −0.8 and −2.2 °C for −3 and −5 °C ambient temperature respectively and had a measured LWC of 2.2 g m$^{-3}$ (Theis et al., 2022; v. Blohn et al., 2009; Pflaum and Pruppacher, 1979). For temperatures higher than −3 °C, no freezing was observed on the Teflon-coated bars and little freezing could be observed on the graupel. During wet growth experiments small cloud droplets coalesced and formed larger mm-sized drops before freezing. A photo of an example ice sample is provided in the Supplement (Figure S1). This coalescence of

droplets during our experiments are representative of the wet growth of graupel rather than the wet growth of hail—where the accreted liquid water is shed off from the surface due to the faster fall speeds.

**2.3 Retention measurements**

Dilute aqueous solutions with different compositions were used for the experiments. Single component measurements were carried out for the α-pinene oxidation products and 2-nitrophenol. The aqueous solution to be analyzed contained one of the

compounds of interest (pinonic acid, pinic acid, pinandiole, or 2-nitrophenol) and sodium bromide (NaBr; Sigma-Aldrich/ ≥99 %). For the other nitro-aromatic compounds (2-nitrobenzoic acid, 4-nitrocatechol, and 4-nitrophenol), a mixture of all organics and NaBr was used, an experimental setup that comes closer to the complex conditions in the atmosphere. All samples contained NaBr as an internal standard (IS) to account for dilution and evaporation effects. The retention of NaBr is assumed to be 1, i.e. NaBr remains completely in the droplets during freezing. HCl (30 %) was added if the measurements were carried

out at a pH value of 4. The concentrations of the chemicals used is shown in Table 1.

**Table 1: Concentrations of the investigated solutions**.

| Compound | Concentration (μmol L$^{-1}$) | Label /purity | IS concentration NaBr (μmol L$^{-1}$) |
|---|---|---|---|
| cis-pinonic acid | 10 | Sigma Aldrich/98 % | 10 |
| cis-pinic acid | 10 | Synthesized/N/A | 10 |
| (1R,2R,3S,5R)-(−)-pinanediol | 15 | Merck/99 % | 10 |
| 2-nitrophenol | 30 | Thermo Scientific/99 % | 10 |
| 2-nitrobenzoic acid | 10 | Thermo Scientific/95 % | 10 |
| 4-nitrocatechol | 10 | Thermo Scientific/≥98 % | 10 |
| 4-nitrophenol | 10 | Alfa Aesar/99 % | 10 |

To generate the supercooled droplets, the solution containing the analyte and the IS was nebulized with a gas stream of nitrogen

(>99.8 %) and either two or four spray nozzles depending on the experiment. Two spray nozzles and a nitrogen flow of



20 L min$^{-1}$ were used for dry growth and four spray nozzles and a flow of 24 L min$^{-1}$ for wet growth conditions. The number of spray nozzles influences the liquid water content (LWC) and thus the growth regime.

For dry growth conditions a lower LWC is required, therefore only two spray nozzles were used. The resulting droplet size distribution in the wind tunnel was measured using a custom in-line digital holographic instrument, similar to the "holographic

imaging and velocimetry instrument for small cloud ice" (HIVIS) described by Weitzel et al. (2020). The measurements were not taken simultaneously to each retention experiment but measured independently under the same conditions during the retention experiments. After the cloud of droplets was produced, a camera (Basler acA2040) captured the holograms containing the images of the droplets at 90 fps. Typical measurement time was 1 minute. Approximately 5400 holograms were reconstructed and analyzed for each measurement with two and four nozzles. Using a telecentric lens with a 0.5 magnification,

the pixel size of the holograms was 2.72 μm x 2.72 μm which yielded a sample area and volume of 5.57 x 5.57 mm$^2$ and 2.48 cm$^3$ respectively. For the reconstruction of the holograms, a depth of 8 cm was chosen as this represents the central part of the measurement section where the collectors were exposed to the cloud of supercooled droplets during experiments. The holograms were reconstructed at each $\Delta z = 100$ μm distance along the optical axis using the method of Fugal et al.(2004). This produced an in-focus image of the droplets independent of their location within the sample volume. The particle sizes

were obtained from a particle detection algorithm of Fugal et al. (2009), which determined the normalized number and volume distributions. Figure 3 (a) shows the number distribution of the supercooled droplets in the measurement section of the tunnel when two spray nozzles were used. Figure 3 (b) shows the volume distribution of the droplets, which represents the normalized cloud liquid water content (LWC) per size interval. We observed no difference in the distribution when using four nozzles (see SI Figure 2). The LWC was measured using a dew-point meter (MBW Calibration Ltd., Wettingen, Switzerland, DP3-D/SH)

in conjunction with a 5 m long heated pipe. Wind tunnel air was sampled isokinetically through the heated pipe to evaporate the droplets. The absolute humidity was then obtained from the dew point measurement. In a second step, a droplet separator was installed at the inlet to the heated tube to measure the dew point of the wind tunnel air without droplets. The difference between the two absolute humidities gives an LWC of $0.9 \pm 0.2$ g m$^{-3}$ for dry growth and $2.2 \pm 0.2$ g m$^{-3}$ for wet growth conditions.




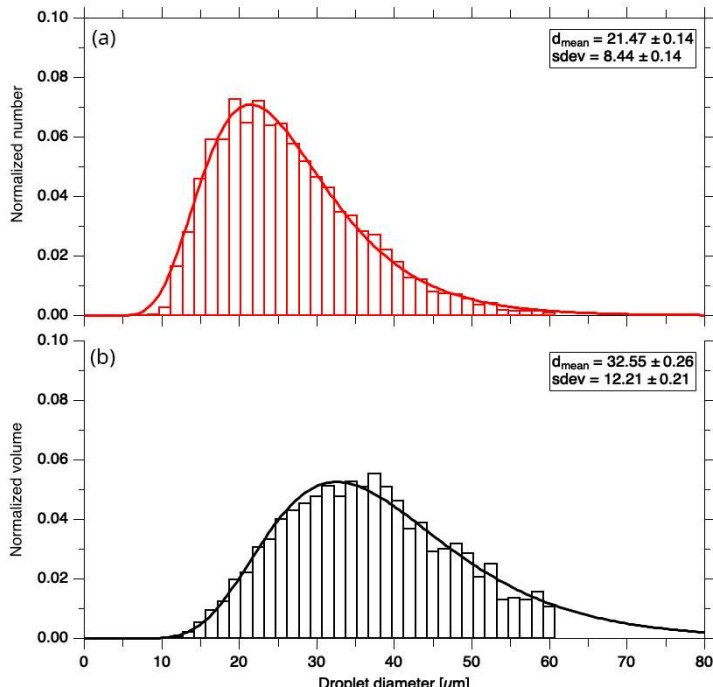


**Figure 3: Normalized droplet number (a) and volume distribution (b) of the supercooled droplets generated using two spraying nozzles. The lines represent log-normal fit functions.**

During retention measurements, the droplets were transported downstream from the sprayer into the experimental section. In this area, the supercooled droplets collided with three different surfaces, which were used as rime ice collectors. The first

surface was a Teflon-coated bar (FEP; outer diameter 6 mm) and the second was an ice sphere (graupel) with a diameter of 7 mm, which comes closer to real atmospheric sizes for graupel. According to the American Meteorological Society glossary of meteorology, graupel are defined as rimed particles with diameters less than 5 mm; larger ones are called hailstones. However, we refer to the investigated particle as graupel according to the flow conditions present rather that of particle size. The larger diameter was necessary to obtain a sufficient sample volume for analysis. The last surface was a cold tube made of

Teflon (PFA) that was constantly filled with liquid nitrogen (LN finger tube). This sample determined the liquid phase concentration of the droplets immediately before riming occurs. At the surface of the LN finger tube, freezing is so fast that the retention can be assumed to be 1 and the concentration of the droplets can be determined before riming.

To produce graupel, a silicon mold was filled with ultra-pure water and frozen. The graupel were "captively floated" to avoid the loss of graupel and any contamination on contact with the wind tunnel walls. For this purpose, they were attached to a

nylon fiber with a diameter of 80 µm. Under these conditions, the rime collectors were exposed to the airflow containing the



supercooled droplets at 3 m/s, which corresponds to a typical fall velocity of a graupel. (Wang and Kubicek, 2013; Pruppacher and Klett, 2010). The ice samples were collected after each experimental run and stored at -25 °C until the chemical analysis.

**2.4 Desorption correction coefficient**

Desorption experiments were carried out to investigate whether the organic compounds introduced into the solution leave the
water droplets by transition into the gas phase after the nebulization process, i.e. whether the compounds are depleted even without a freezing process We tested 2-nitropheno, because this species has the lowest effective Henry's law constant and therefore the highest probability of transition from the droplet phase to the gas phase, resulting in a high gas phase concentration. The same solution (see Table 1) used for the retention measurements, containing 2-nitrophenol and NaBr as IS, was nebulized and the droplets were transported downstream in the wind tunnel. The droplets were sampled in a vial after a
residence time of approximately 2 s in the tunnel. The solution was then collected and measured by UHPLC-HRMS. Desorption measurements were carried out at nebulization nitrogen flow rates between 20 and 24 L min$^{-1}$ and different temperatures (0 °C, 5 °C, 10 °C, and 17 °C) to obtain a temperature dependency. This was then extrapolated to the temperatures and flow rates used in the retention experiments.

**2.5 UHPLC-HRMS analysis**

For analysis, the ice samples were melted and filtered through polyamide (PA) membranes (pore size: 0.20 µm; Altmann Analytik). Analysis was performed in triplicate using a Dionex UltiMate 3000 ultra-high-performance liquid chromatography (UHPLC) system coupled to a heated electrospray ionization source (HESI) or atmospheric pressure chemical ionization (APCI), and a high-resolution Q-Exactive Orbitrap mass spectrometer (HRMS) (all Thermo Fisher Scientific). A Hypersil Gold, C18, 50 x 2.0 mm column with 1.9 µm particle size (Thermo Fisher Scientific) was used for the chromatography. Eluent
A consisted of 98 % LC/MS grade water (Thermo Fisher Scientific) with 0.04 % formic acid and acetonitrile (VWR Chemicals), eluent B consisted of 98 % acetonitrile and water, and the injection volume was 5 µL. Different $H_2O$/ACN gradients were used for the different compounds. For pinonic acid, pinic acid, pinandiole, 2-nitrobenzoic acid, 4-nitrocatechol, and 4-nitrophenol, a flow rate of 0.5 mL min$^{-1}$ and a gradient as described following were used: Starting with 2 % B isocratically for 1 min, increasing to 20 % B in 2.5 min, then further increasing to 90 % in 1.5 min, after which B was held at
90 % for 4 min, decreased to 2 % in 0.5 min, and held again for 1.5 min. For pinanediol, a post-column flow of 50 mmol L$^{-1}$ $NH_4OH$ in MeOH was added after 1 min at a flow rate of 0.1 mL min$^{-1}$ to enhance ionization. The HESI source was used in negative mode, resulting in the formation of deprotonated molecular ions. Sheath gas and auxiliary gas flow was 40 and 20 a. u. respectively. The temperature of the auxiliary gas heater was 150 °C and the capillary temperature was 350 °C. The sprayer voltage was set to -4.00 kV.

A different gradient with a flow rate of 0.3 mL min$^{-1}$ was used for 2-Nitrophenol. Starting with 2 % B isocratically for 1 min, increasing to 20 % B in 2.5 min, then further increasing to 90 % in 1.5 min, after which B was again held at 90 % for 0.3 min, decreased to 2 % in 0.2 min and held again for 0.5 min. To further enhance ionization, a post-column flow of 50 mmol L$^{-1}$



NH$_4$OH in MeOH was added after 1 min at a flow rate of 0.1 mL min$^{-1}$. The APCI source was used in negative mode, resulting in the formation of deprotonated molecular ions. Sheath gas and auxiliary gas flow was 23 and 5 a. u., respectively. The

vaporizer temperature was 375 °C and the capillary temperature was 350 °C.

### 2.6 Calculation of the retention coefficient

Eq. (1) was used for compounds with a retention coefficient $R$ close to 1:

$$R = \frac{c_{\text{compound}}^{\text{sample}} / c_{\text{compound}}^{\text{LN}}}{c_{\text{IS}}^{\text{sample}} / c_{\text{IS}}^{\text{LN}}} \tag{1}$$

The numerator describes the ratio between the concentration of the compound of interest in the ice sample ($c_{\text{compound}}^{\text{sample}}$) (Teflon

coated bars or graupel) and in the LN finger sample ($c_{\text{compound}}^{\text{LN}}$). The denominator describes the same ratio but for the IS

($c_{\text{IS}}^{\text{sample}}$ / $c_{\text{IS}}^{\text{LN}}$). Thus, it is not necessary to consider a dilution, evaporation, and desorption correction as these effects change

both the compound and IS concentration in the nitrogen finger sample accordingly. Therefore, a change in this ratio is solely

an effect of the retention of the compound during the riming process.

The calculation is different for compounds with a lower retention coefficient. Because these compounds are transferred to the

gas phase in larger amounts before and during the freezing process, a higher gas phase concentration is present in the tunnel.

Therefore, it cannot be ruled out that the measured concentrations of the LN finger tube samples could be influenced by

additional adsorption out of gas phase components. This would no longer provide a suitable correction for determining the

retention coefficient.

As the distance between the two sampling points (nebulizer and graupel/bar) is large, it is necessary to determine the amount

of compound that is transferred to the gas phase before freezing. A desorption correction coefficient $D_{\text{compound}}$ (Eq. (2)) is

introduced for this purpose. The definition of the desorption correction coefficient is analogous to that of the retention

coefficient (Eq. (1)), i.e. the numerator also describes the ratio of the compound remaining in the sample (droplets sampled in

a vial) to a reference sample. However, in this case, the ratio is not relative to LN finger tube sample but to the sprayer solution,

which is the solution immediately before droplet formation, takes place ($c_{\text{compound}}^{\text{sample}}$ / $c_{\text{compound}}^{\text{sprayer}}$). The denominator again

describes the same ratio but for the IS ($c_{\text{IS}}^{\text{sample}}$ / $c_{\text{IS}}^{\text{sprayer}}$):

$$D_{\text{compound}} = \frac{c_{\text{compound}}^{\text{sample}} / c_{\text{compound}}^{\text{sprayer}}}{c_{\text{IS}}^{\text{sample}} / c_{\text{IS}}^{\text{sprayer}}} \tag{2}$$

To calculate the retention coefficient $R$ for compounds with a lower retention coefficient, Eq. (1) was extended to include the

desorption correction coefficient using the sprayer solution sample instead of the LN finger (see Eq. (3)):

$$R = \frac{c_{\text{compound}}^{\text{sample}} / c_{\text{compound}}^{\text{sprayer}}}{c_{\text{IS}}^{\text{sample}} / c_{\text{IS}}^{\text{sprayer}} \cdot D_{\text{compound}}} \tag{3}$$





**3 Results and discussion**

**3.1 Desorption correction coefficient**

Since different nitrogen flow rates for dry and wet growth conditions were used for nebulization in the retention experiments, the desorption of 2-nitrophenol was determined for the different gas flow rates. To maintain the cloud of droplets with four nozzles, the nebulizer gas flow rate had to be increased. Figure 3 shows that there is a strong dependence of the desorption

correction coefficient of 2-nitrophenol on temperature, indicating less desorption at lower temperatures. This is expected since the mass transfer rate of a dissolved compound to the environment decreases for decreasing temperatures. The reason for this is that the mass transfer depends, among others on the temperature-dependent parameters like gas- and aqueous-phase diffusivities and the effective Henry's law constant. The desorption coefficient for four sprayers is significantly reduced compared to the two-sprayer case. This is attributed to the slightly longer exposure time in the wind tunnel.

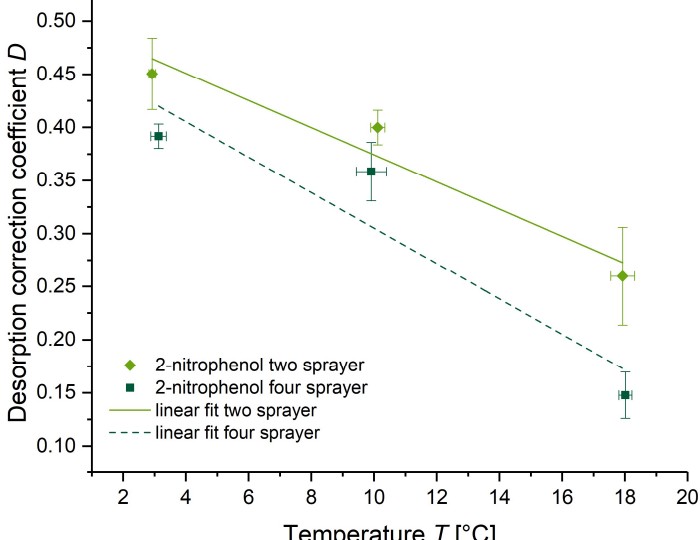

**Figure 4: Desorption correction coefficient for 2-nitrophenol with a nitrogen flow for nebulization of 20 L/min (light green diamonds) and 24 L/min (dark green squares). The errors correspond to the standard deviation of each of the 8 to 10 desorption measurements at the corresponding temperatures. The lines represent the linear regression for each set of measurements.**

A linear trend is visible for all measurement series. A linear regression with the equation $D_{compound} = a_D \cdot x \cdot [°C]^{-1} + b_D$ was performed for all of them to extrapolate the desorption to the temperatures present during the experiments. These results

are listed in Table 2. Extrapolating the results for two sprayers to the experimental conditions (e.g. -10 °C and -4 °C) yields $D = 0.63$ and $D = 0.55$ for typical dry and wet growth conditions respectively. These corrections to the retention coefficient due to desorption are significant.





**Table 2: Slope and y-axis intercepts of the linear regressions of the desorption data for the different nitrogen flows.**

| Number of sprayers | Nitrogen flow / $L \cdot min^{-1}$ | $a_D$ / $L \cdot min^{-1}$ | $b_D$ | $R^2$ |
|---|---|---|---|---|
| 2 | 20 | $-0.013 \pm 0.003$ | $0.502 \pm 0.038$ | 0.9428 |
| 4 | 24 | $-0.017 \pm 0.006$ | $0.472 \pm 0.072$ | 0.8850 |

**3.2 Retention measurements**

The results of the retention measurements for the α-pinene oxidation products are shown in Figure 5; those for 4-nitrocatechol, nitrobenzoic acid, and 4-nitrophenol are shown in Figure 6; and the desorption corrected $R$ for 2-nitrophenol in Figure 7.

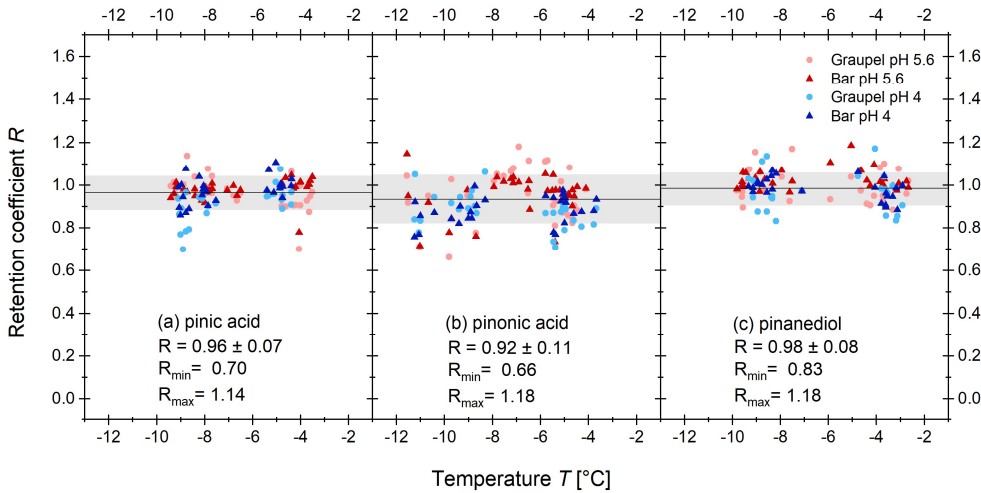

**Figure 5: Experimentally determined retention coefficients of (a) pinic acid, (b) pinonic acid, and (c) pinanediol as a function of the temperature during the experiment for different rime collectors. The circles represent the graupel samples and the triangles those**
**of the Teflon-coated bars. For the blue symbols, the pH was adjusted to 4 by adding HCl (30 %), the red symbols are without adding HCl. The solid black line represents the mean of all measurements, and the grey area represents one standard deviation.**

**3.2.1 Pinic acid and pinonic acid**

Figure 5 shows the results for (a) pinic and (b) pinonic acids. Pinic acid and pinonic acid both show neither a pH or temperature dependence nor influence from the rime regime. This agrees with the earlier findings for compounds with high $H^*$. The
retention coefficients for pinic acid vary between 0.65 and 1.14 with a mean value of $0.96 \pm 0.08$. For pinonic acid, a mean retention coefficient of $0.92 \pm 0.10$ was determined. The variation in the measured values of $R$ is greater than that for pinic acid and lies between 0.66 and 1.18. This is probably due to the lower $H^*$, which makes pinonic acid more sensitive to slight changes in the experimental conditions.



### 3.2.2 Pinanediol

The results for pinanediol are depicted in Figure 5 (c). No effect on the investigated parameters could be found for this diol. The mean retention coefficient was $0.98 \pm 0.08$ with a minimum value of 0.83 and a maximum value of 1.18, which is an exception to previous findings (Jost et al., 2017). In comparison to pinic and pinonic acid, pinanediol showed a lower $H^*$ (see Table 4). A lower retention coefficient is expected for compounds in this range as well as a dependence on temperature. However, no temperature, pH, or growth regime dependence was seen. This deviation from the expected behavior could be

because the Henry's law constant for pinanediol is estimated using the bond method implemented in the HENRYWIN$^{TM}$ software as part of EPI Suite$^{TM}$ The bond method breaks down the molecule into a sum of the individual bonds that make up the compound. The exact structure and spatial orientation of the molecule is not considered. Due to the large deviations of the polyols in the original method by Hine and Mookerjee (1975), further correction factors were included in the method. However, the predictions are less accurate for molecules with a more complex structure (Meylan and Howard, 1991; Hine and Mookerjee,

1975). Pinanediol is a cyclic diol. Due to its capped ring form, it is sterically hindered and thus the potential interactions between the OH groups are not considered by the model. Since there is no experimental data for the Henry's law constant, there is no current alternative to the estimation.

### 3.2.3 4-Nitrocatechol and 4-nitrophenol

The results for 4-nitrocatechol and 4-nitrophenol are shown in Figure 6 (a) and 6 (b). Overall, no pH or temperature dependence

is recognizable. Furthermore, no difference between wet and dry growth conditions can be observed. This agrees with the current literature. As Stuart and Jacobson (2004, 2003) have shown, a temperature or pH dependence is not expected for compounds with high effective Henry's law constants, such as 4-nitrocatechol or 4-nitrophenol. A mean retention coefficient of $1.01 \pm 0.14$ and $1.01 \pm 0.07$ was determined for 4-nitrocatechol and 4-nitrophenol respectively. For 4-nitrocatechol, the derived retention coefficients showed a large scatter that has no current explanation. A minimum value of 0.63 and a maximum

value of 1.41 was obtained. 4-nitrophenol showed a smaller scatter with values between 0.79 and 1.28.



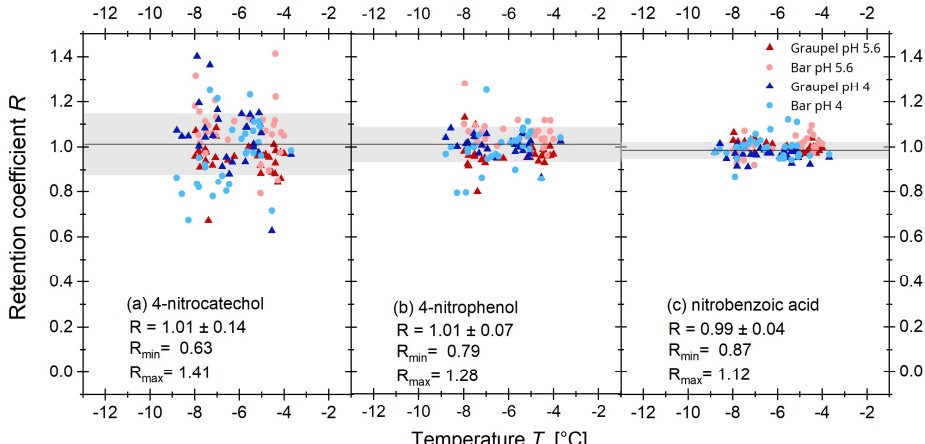

**Figure 6: Experimentally determined retention coefficients of (a) 4-nitrocatechol, (b) 4-nitrophenol, and (c) nitrobenzoic acid as a function of the temperature during the experiment for different riming collectors. The circles represent the graupel samples and the triangles those of the bars. For the blue symbols, the pH was adjusted to 4 by adding HCl (30 %), the red symbols are without adding**

**HCl. The black line represents the mean of all measurements, and the grey area represents one standard deviation.**

### 3.2.4 2-Nitrobenzoic acid

For 2-nitrobenzoic acid in Figure 6 (c), no dependence on the investigated parameters was found. The average retention coefficient is $0.99 \pm 0.04$, with the lowest scatter among the measured compounds. The smallest value measured was 0.87 and the largest 1.12. This is probably because 2-nitrobenzoic acid has the lowest pKa value of all the compounds studied. In the

pH range investigated, most of the 2-nitrobenzoic acid molecules are deprotonated, unlike the other components studied. This makes a transition into the gas phase less likely and therefore small differences in the measurement conditions have less influence on the results.

### 3.2.5 2-Nitrophenol

2-Nitrophenol (Figure 7) showed significantly lower retention coefficients than the other compounds presented above. The

different rime collectors (Teflon-coated bar, graupel) at different growth conditions (pH 4 and 5.6) showed a statistically significant (significance level $\alpha = 0.05$) negative temperature dependence (lines and equations in Figure 7; Table 3). It is noticeable that the slopes for the Teflon coated bar samples for pH 4 and 5.6 are equal within the error range. The same applies to the graupel samples. The bar samples show a slightly stronger temperature dependence than the graupel samples. This is probably due to the metal rod inside the Teflon-coated bar and the resulting faster heat transfer. It is also noticeable in these

measurements that the graupel samples (light red and light blue circles in Figure 7 (b)) have an overall lower retention



coefficient than the bar samples (red and blue triangles in Figure 7 (a)). This might also be explained by the metal rod inside the Teflon-coated bar. The faster heat transfer leads to a shorter freezing time and thus to a higher retention coefficient.

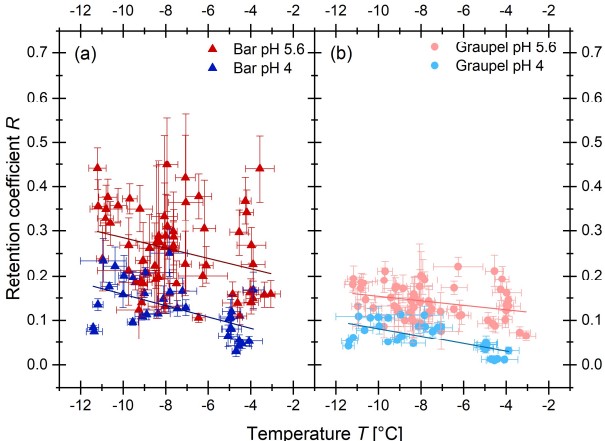

**Figure 7: Experimentally determined retention coefficients of 2-nitrophenol as function of the temperature during the experiment for different rime collectors. The triangles represent the Teflon coated bar samples (a) and the circles those of the graupel (b). For the blue symbols, the pH was adjusted to 4 by adding HCl (30 %), the red symbols are without adding HCl. The line represents a linear fit.**

The data points (Figure 7) below −6 °C were obtained from dry growth conditions, while the ones higher than that represent wet growth conditions. A closer inspection of the data reveals that the temperature dependency might also be a result of the riming regimes. When separating the data between the two regimes the temperature dependency vanishes. Therefore, the difference in retention values between wet and dry growth conditions might be due to the longer freezing times of the accreted and coalesced droplets. The droplets accreted during wet growth regime remained liquid for some time and formed a larger mm-size drop before they froze. This means dissolved 2-nitrophenol could further desorb from the drops before ice shell formation occurred, which reduces further expulsion from the freezing droplet. This resulted in slightly lower retention values for the rime collectors grown during wet growth. Considering the absolute retention values for wet growth, it becomes apparent that 2-nitrophenol will not retain in atmospherically significant amounts during wet growth conditions. However, as the values between dry and wet growth overlap within the experimental uncertainty, a temperature dependency is provided here, which is applicable within the investigated temperature and growth conditions range simulated in the present study. As the temperature dependence in the analyzed range has only a minor influence on the retention compared to the data scattering, the mean values were determined for all sets of measurements. The mean retention coefficients for the various riming conditions are shown in Table 3. The overall mean retention coefficient was determined to be $0.16 \pm 0.10$. The measured values of $R$ were between a minimum of 0.01 and a maximum of 0.44.



In addition to the dependence on temperature, growth regime, and the riming collectors already discussed, the pH dependence was also analyzed as retention is also strongly dependent on the dissociation of the molecules. As the retention for the rime

collectors is different, the comparisons are only carried out within the same collectors. The mean retention coefficients for the graupel samples at pH 5.6 and pH 4 were $0.14 \pm 0.04$ and $0.06 \pm 0.03$ respectively. At pH 5.6 it is more likely that 2-nitrohenol is dissociated in comparison to pH 4. A dissociated molecule needs to be neutralized by a proton before leaving the droplet during riming. At pH 5.6, more molecules are available that do not evaporate without recombination which might explain the higher retention coefficient.


Table 3: Measured retention coefficient of 2-nitrophenol and their temperature dependencies.

| Conditions | Mean retention coefficient $R$ | Temperature dependency of $R$ |
|---|---|---|
| **Bar pH 5.6** | $0.26 \pm 0.09$ | $R_{B_{5.6}} = (-0.011 \pm 0.005)\,T + (0.171 \pm 0.038)$ |
| **Graupel pH 5.6** | $0.14 \pm 0.04$ | $R_{G_{5.6}} = (-0.005 \pm 0.002)\,T + (0.104 \pm 0.017)$ |
| **Bar pH 4** | $0.13 \pm 0.06$ | $R_{B_4} = (-0.013 \pm 0.004)\,T + (0.032 \pm 0.028)$ |
| **Graupel pH 4** | $0.06 \pm 0.03$ | $R_{G_4} = (-0.008 \pm 0.002)\,T + (-0.004 \pm 0.014)$ |

### 3.3 Relationship between Henry's law constant and retention

Jost et al. (2017) have shown a correlation between the retention coefficient $R$ and the dimensionless effective Henry's law

constant $H^*$ ($H^* = K_H \cdot \bar{R}T$, with $K_H$: effective Henry's law constant; $\bar{R}$: gas constant; $T$: temperature) for inorganic and small organic molecules, which can be described by Eq (4).

$$R(H^*) = \left(1 + (a\,/\,H^*)^b\right)^{-1}, \tag{4}$$

where $R$ is the retention coefficient, $H^*$ is the dimensionless effective Henry's law constant and $a$ and $b$ constants that are determined by a fit. To test whether this relationship is also applicable to the larger organic molecules investigated here, the

mean values of retention coefficients of the compounds were plotted against their dimensionless effective Henry's law constant (Figure 8). $H^*$ was determined for the investigated pH values and at 298 K. Since most of the compounds analyzed in this study did not show any pH dependence, a pH value of 4 was used for the calculation of $H^*$ for these compounds. This is a typical value found in cloud water samples (Pye et al., 2020; Löflund et al., 2002). For 2-nitrophenol, the mean retention coefficient for each of the pH values (pH 4 and 5.6) were calculated and plotted (Figure 8). The calculated values of $H^*$ are

listed in Table 4. Since there are no measured Henry's law constants nor reaction enthalpies for some of the more complex organic compounds, these were predicted using the HENRYWIN™ software which provides the values for 298 K.



**Table 4: Retention coefficient of the measured compounds, their pK$_a$ values, Henry's law constants, and dimensionless effective Henry's law constants at 298 K and pH 4 for all compounds except 2-nitropheand (pH 4 and 5.6).**

| Compound | Mean retention coefficient $R$ | Acid dissociation constant pK$_a$ | Henry's law constant[1] / M/atm | dimensionless effective Henry's law constant |
|---|---|---|---|---|
| 4-nitrocatechol | $1.01 \pm 0.14$ | $7.23$[2] | $4.35 \cdot 10^9$ | $1.06 \cdot 10^{11}$ |
| 2-nitrobenzoic acid | $0.99 \pm 0.04$ | $2.17$[3] | $2.08 \cdot 10^6$ | $3.49 \cdot 10^9$ |
| cis-pinic acid | $0.96 \pm 0.07$ | $4.64$[2] | $1.02 \cdot 10^8$ | $3.06 \cdot 10^9$ |
| cis-pinonic acid | $0.92 \pm 0.11$ | $5.19$[4] | $1.12 \cdot 10^6$ | $2.93 \cdot 10^7$ |
| 4-nitrophenol | $1.01 \pm 0.07$ | $7.15$[3] | $4.52 \cdot 10^5$ | $1.11 \cdot 10^7$ |
| (−)-pinanediol | $0.98 \pm 0.08$ | $14.68$[2] | $4.08 \cdot 10^3$ | $9.97 \cdot 10^4$ |
| 2-nitrophenol (pH 5.6) | $0.20 \pm 0.09$ | $7.23$[3] | $1.43 \cdot 10^2$ | $3.50 \cdot 10^3$ |
| 2-nitrophenol (pH 4) | $0.09 \pm 0.06$ | $7.23$[3] | $1.43 \cdot 10^2$ | $3.58 \cdot 10^3$ |

[1] Calculated using US EPA. [2012]. Estimation Programs Interface Suite™ for Microsoft® Windows, v 4.11 or insert version used]. United States Environmental Protection Agency, Washington, DC, US; [2] Calculated using Advanced Chemistry Development (ACD/Labs) Software V11.02 (© 1994-2024 ACD/Labs); [3] (Haynes, 2014); [4] (Kołodziejczyk et al., 2019).

The compounds measured in this study are the colored, filled symbols in Figure 8; the ones measured by Jost et al.(2017), v.
Blohn et al. (2013; 2011) and in earlier investigations in the wind tunnel are the grey, half-filled symbols. The grey dashed line represents the fit for the grey, half-filled symbols, i.e. previous measurements. Since the retention of formaldehyde, as shown in Jost et al. (2017), depends not only on $H^*$ but also on the hydration to the diol, the value is not taken into account when determining the fit. The resulting parameters are $a_{\text{grey}} = (2.41 \pm 1.06) \cdot 10^4$ and $b_{\text{grey}} = 0.27 \pm 0.04$. The red solid line is the new fit function in which all compounds are considered. Here again formaldehyde was not included into the
fit. Pinanediol was also excluded from the fit as it appears to be an outlier (see Sect. 3.2.2). The values $a_{\text{red}} = (3.88 \pm 2.19) \cdot 10^4$ and $b_{\text{red}} = 0.37 \pm 0.07$ result for the fit function.



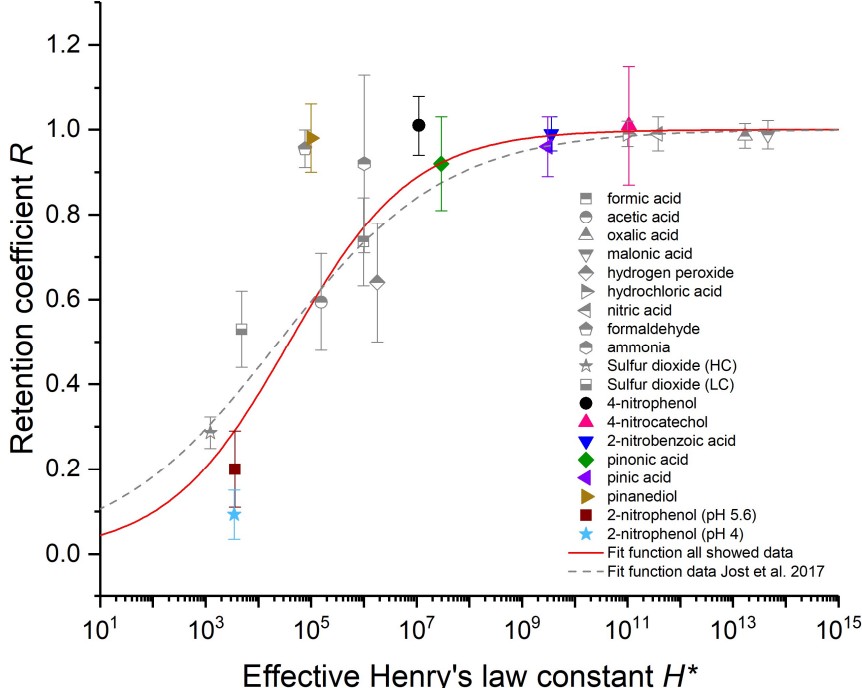

**Figure 8: Measured retention coefficient as a function of $H^*$. Colorful filled symbols: compounds investigated in the present study.**
**Grey symbols: wind tunnel data from earlier studies (Jost et al., 2017; v. Blohn et al., 2013; 2011) Red solid line: new fit to wind**
**tunnel data. Grey dashed line: fit of the grey data points only.**

As Figure 8 shows, the retention coefficients of larger organic molecules do also scale with $H^*$ which agrees with the previous

measurements. Only pinanediol and formaldehyde do not fit at first glance. A comparison of the new fit function (red solid

line) with the values already known from the literature (grey dashed line) shows that the inflection in the fit function is steeper

than previously assumed. Our results show that the retention of compounds with an $H^*$ below $10^3$ is close to 0, i.e. the entire

amount of the compound dissolved in water is released into the gas phase during riming. This is also evident from the

equilibrium distribution of species between the liquid and gas phase in a confined system, which is a function of the LWC (see

Supplement Figure S3). For species with $H^* < 10^3$ just a maximum of 10 % is present in the liquid droplets even for LWCs of

up to 10 g m$^{-3}$, which are only reached within severe storms (Lohmann et al., 2020). For $H^* > 10^3$ the compounds are present

in significant amounts. For compounds with an $H^*$ value above $10^8$, a retention of 1 is expected and the compound remains

completely in the ice phase during freezing. For the compounds in-between $10^3$ and $10^8$, the retention coefficient is highly

dependent on $H^*$. However, the retention estimation is still afflicted with some degree of uncertainty especially in this $H^*$



range. A close inspection of the data shows that it is not obvious that pinanediol and formaldehyde are outliers. It might also be that the transition from low to high retention values occurs in a smaller range of $H^*$ and retention is 1 already for $H^* > 10^5$.

We cannot ultimately clarify the steepness of the curve and the limiting regimes might be closer than assumed here. Jost et al. (2017) argued that formaldehyde cannot be explained solely by $H^*$ and aqueous phase kinetics must be considered. However, such explanations are lacking for pinanediol. This might indicate that the transition from low to high retention values occurs at lower $H^*$. Support for that is given from the fit when taking all measurements into account, i.e. also formaldehyde and pinanediol (see Supplement Figure S4). To finally answer the question about the transition behavior of the function, further

retention measurements in the range $10^4 < H^* < 10^6$ are needed. However, to our current knowledge, and particularly because of the lack of reliable measurements of $H^*$ for pinanediol, we believe that the parameterization given above (Eq. (4), with $a_{\text{red}} = (3.88 \pm 2.19) \cdot 10^4$ and $b_{\text{red}} = 0.37 \pm 0.07$ is most trustworthy to calculate retention in the transition regime.

**4 Conclusions**

Wind tunnel experiments were carried out in the vertical wind tunnel of the Johannes Gutenberg-University Mainz to determine

the retention coefficients $R$ of three α-pinene oxidation products and four nitro-aromatic compounds. The experiments were performed in the temperature range from -12 to -3 °C with a liquid water content of $0.9 \pm 0.2\,\text{g m}^{-3}$ and $2.2 \pm 0.2\,\text{g m}^{-3}$ to represent dry and wet growth conditions and to simulate mixed-phase cloud conditions. The wind speed ($3\,\text{m s}^{-1}$) was chosen to match the typical fall velocity of graupel. The temperature and pH dependences (pH 4 and 5.6) of the compounds were studied as well as the dependence on the growth regimes. Two different rime collectors were also considered, a Teflon-

coated metal bar and a graupel. Stuart and Jacobson (2004; 2003) hypothesized a dependence on external influences such as temperature and pH for compounds with a low dimensionless effective Henry's law constant $H^*$. Of the compounds investigated, 2-nitrophenol has the lowest $H^*$. In agreement with the current literature, the determined retention coefficients of 2-nitrophenol show a significant dependence on pH and on the type of rime collector. The mean retention coefficients for the graupel samples are $R_{\text{G5.6}} = 0.14 \pm 0.04$ (pH 5.6) and $R_{\text{G4}} = 0.06 \pm 0.03$ (pH 4). This indicates that changes in the pH

value can influence retention and must be taken into account. 2-nitrophenol also showed a dependence between dry and wet growth conditions. The temperature dependence and the dependence on growth conditions seem to overlap, as the growth conditions differ not only in liquid water content but also in temperature. Since the measured values between dry and wet growth overlap within the experimental uncertainty, a temperature dependence is reported in this study.

2-Nitrobenzoic acid, 4-nitrophenol, and 4-nitrocatechol showed retention coefficients of $0.99 \pm 0.04$, $1.01 \pm 0.07$, and

$1.01 \pm 0.14$, respectively, with no significant dependence on temperature, pH, type of rime collector or growth-regime. For cis-pinic acid and cis-pinonic acid, retention coefficients of $0.96 \pm 0.07$ and $0.92 \pm 0.11$ were obtained, which also showed no dependence on the parameters investigated. This study shows that there appears to be no difference between dry and wet growth conditions for compounds with a high effective Henry's law constant and that $H^*$ can also be used to estimate retention coefficients for wet growth conditions, at least for graupel. This is in contrast to a modelling study by Michael and Stuart



(2009), which indicates a lower influence of $H^*$ and low retention coefficients even for compounds with high $H^*$ under wet growth conditions for hailstones. The retention coefficient for pinanediol was determined to be $0.98 \pm 0.08$ with no significant temperature dependence. This would not have been expected due to the comparably low $H^*$. However, the Henry's law constant is only predicted and may be subject to errors due to the specific structure of the molecule. For this reason, it is important to determine Henry's law constants of more different molecules to aid in the understanding and modeling of processes in the

atmosphere.

The retention coefficients for 4-nitrophenol and 2-nitrophenol differ considerably. Since they are structural isomers, it is obvious that they have the same molecular formula and the same functional groups, although they are arranged differently. The group method of the HENRYWIN$^{TM}$ software predicts the same Henry's law constant. This clearly shows the importance of reliable prediction or measurement of $H^*$ and the importance of chemical structure.

It is demonstrated here that the retention coefficients of more complex organic molecules depend mainly on the dimensionless effective Henry's law constant. These experiments have improved the parameterization of retention coefficients using the dimensionless effective Henry's law constant. The results show that the retention of compounds with an $H^*$ below $10^3$ is negligible and thus the entire amount of the compound dissolved in the supercooled drops is released into the gas phase during freezing. For compounds with an $H^*$ value above $10^8$, a retention of 1 is expected and the compound remains completely in

the ice phase during freezing. These compounds can be effectively washed out by precipitation or transported further upwards and released by sublimation of the ice particles. At high altitudes and low temperatures, the volatility of these compounds is even lower, and it is possible that particulate residuals form after sublimation. This probably has an impact on the chemistry of the upper troposphere and ultimately on the Earth's radiative budget. In the intermediate range of $H^*$, the improved fit of $R$ vs $H^*$ (Eq. (4), with $a_{\mathrm{red}} = (3.88 \pm 2.19) \cdot 10^4$ and $b_{\mathrm{red}} = 0.37 \pm 0.07$) can be used to estimate the retention coefficient

and thus further improve cloud models that account for transport of organic trace components. However, the present results show that more retention measurements are needed for compounds with $10^4 < H^* < 10^6$ to clarify the sharpness of the transition between the two boundaries of zero retention and full retention.

Author contribution

CB, JS, MG, KD, YM, AA, LG, AT, AV, TH designed and performed the wind tunnel experiments; FU synthesized pinic acid; CB performed the analytical measurement, analyzed the data and wrote the manuscript draft; JS, MG, MS; AT; AV; TH reviewed and edited the manuscript.

Competing interests: The contact author has declared that none of the authors has any competing interests.


Acknowledgements: This work was funded by the Deutsche Forschungsgemeinschaft (DFG, German Research Foundation) – TRR 301 – Project-ID 428312742.



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
