# Peer review of "Retention of α-pinene oxidation products and nitro-aromatic compounds during riming"

_EGUsphere, 2024_

## Referee Comment (RC1)

**"Retention of Pinene Oxidation Products and Nitro-aromatic Compounds during Riming"**

**By C. Borchers et al. (2024) for EGUsphere**

Reviewed by Jeff Snider, University of Wyoming

The manuscript adds to a body of measurements of retention during riming. A retention coefficient increases from zero to one in the limit that volatilization to the gas phase does not occur during riming. In that limit the material is scavenged from the gas phase, via its incorporation in a graupel particle, opening the possibility for vertical transport and removal.

The measurements evaluated in this contribution can help improve understanding of new particle formation (NPF).

More broadly, the investigation casts a spotlight on parameterizing retention in terms of Henry Law solubility.

The authors should consider my critiques and reply with a revised manuscript.

**L36-37**

As scientists we are striving to better understand tropospheric chemistry - the associated roles of aerosol and cloud processes – while aiming to reliably model what is happening. That is clear. However, the Introduction seems overly focused on the upper troposphere and on organic compounds. What is depicted in Figure 1 is also important for latitudes other than tropical and for compounds other than IVOC and SVOC. Sulfur dioxide and sulfuric acid fall into the camp of compounds not mentioned at this point in the Introduction. My recommendation is that you adjust somewhat, so that readers are not left with the impression that the motivating uncertainty is only scenarios in deep convective clouds, or in tropical deep convective clouds, or that the uncertainty only applies to the cloud processing of organic compounds.

**L32-35**

The statement that tropical convection does not produce sinks for small particles (and condensible vapors) needs clarity.  If sinks (aerosol surface ?) are missing, then this can accelerate NPF. But, if tropical convection also removes gaseous precursor, then NPF is decelerated.  In this context, what is known about non-tropical convection?

**L53 – 54**

It's not clear what is implied by "autoxidation."

**L85 – 88**

In those prior investigations, were drops or droplets collected from regions that were _not _ supercooled?  If that was the case, then this statement is not obviously true.

**L103**

Is the "simulated graupel" here the same at the "captively floated" target discussed later?

**2.2 Growth Regimes**

During wet growth, broadly speaking, the sample is at ~ 0 °C, droplets are collected, some of that material adds to the mass of the sample, and some is shed. Your observation is that T > -3 oC (this is an ambient temperature threshold, correct?) make for "no freezing." Could this be because the simulated graupel (and the bar) are thermally coupled to a warmer apparatus?

A comment: Saying that the layer is freezing "very slowly" is confounding an already difficult concept. I will argue that, during wet growth, the freezing rate of an element of input liquid is impossible to calculate. In contrast, during dry growth, freezing rate can be calculated because shape, mass, and boundary conditions are constrained. Rates are fast (the impacted droplet is small, and the temperature gradient is reasonably large) and the characteristic time is small (<< 1 s).

I like how you have tied with the theoretical work of Michael and Stuart (2009) and brought in your observations of impinged droplets forming larger surface elements. What is the evidence that there is no shedding?

**L160 –**

You present _normalized_ number and _normalized_ mass distributions. Why can't this method be used to quantify LWC?

**L174 –**

Can you reference a thesis, dissertation, or publication where the distance between the rime collector (s) and the sprayer is documented? If not, please specify that distance.

**L180 – 182**

This needs better clarity. The apparatus captured droplets on an impaction substrate where they froze to form rime. Subsequently you melted the sample and measured the concentration of analyte in the liquid. Please revise for clarity.

**L191 –**

There is a typo in this sentence. The same typo is on L348.

**L200 –**

Why were the melted samples filtered? Is it possible that analyte (or IC) was lost or gained during this step?

**L212 –**

What is "a. u."? This occurs twice in this Section.

**L234 –**

The sentence has "nebulizer" but Figure 2, and the equations, have "sprayer." These are the same thing, I believe. "Nebulizer" and "nebulization" also appear later in the manuscript. Please use consistent terminology.

**L248 –**

It is important to know if the other nitrophenol isomer, the one with the larger $K_H$, was tested for desportion.

What about other $K_H < 10^5$ M atm$^{-1}$ compounds? For example, was the desportion of pinanediol tested? These $K_H < 10^5$ M atm$^{-1}$ compounds partition mostly into the gas phase, assuming phase equilibrium.

**L255 –**

Longer exposure time in the wind tunnel, with two sprayers compared to four sprayers, is consistent with the discussion of nitrogen volumetric flow rates. With four sprayers, the nitrogen flow rate is larger, and so, the exposure time is smaller. Based on that, I'm expecting longer exposure time with two sprayers, more liquid-to-gas desportion with two sprayers, and a smaller "desportion correction coefficient" with two sprayers. Larger "desportion correction coefficient", with two sprayers, is contrary to Figure 4. Because the fit lines are converging toward smaller temperature, this may not matter. But it's important to get the interpretation of Figure 4 correct.

**L286 –**

A period is missing.

**L283 -**

How is your result, for pinanediol, an exception to that reported in Jost et al. (2017)? Chemically, the pinanediol (1, 2) is structurally different from the formaldehyde diol.

**Table 4 and its footnotes –**

The footnotes, and associated citations, are obscuring this aspect of the presentation. Why can't the US EPA citation be "US EPA, 2012"? Related to this, please see my comment below about the two Henry constants you present (Table 4) for the two structural isomers.

**L436 –**

The 2 nitrophenol isomer may form an intramolecular hydrogen bond, possibly weakening its binding to solvent (water). This same rationale might also explain why the Henry constants are so markedly different. There is some discussion of this in the chemical literature. Please provide some insight, and reference citations, in your revision.

**L438 (and Table 4) –**

It is not clear how you arrived at the different Henry constants for the two structural isomers. Did these values come from HENRYWIN? Here (L438) you state that the two isomers are predicted to have the same $K_H$. Also (L365), you state that "The calculated values of H* are listed in Table 4. Since there are no measured Henry's law constants nor reaction enthalpies for some of the more complex organic compounds, these were predicted using the HENRYWIN software which provides the values for 298 K." This is confusing.

---

## Referee Comment (RC2)

**Borchers et al. "Retention of a-pinene oxidation products and nitro-aromatic compounds during riming"**

General comments

This is an interesting and important experimental study of retention during riming that supports, refines, and expands the existing knowledge on the phenomenon. It further contributes to understanding and parameterization of the fate of a variety of compounds in freezing clouds, with potential implications for understanding of particle formation and other chemistry of the troposphere. The experimental approach is clever and carefully includes measurement of conditions and properties that have been hypothesized to affect retention, but for which more experimental data are needed. However, some of the calculation methods are unclear and need to be more rigorously explained (or applied), particularly regarding the use of the desorption/absorption correction and its extrapolation. Additionally, the conclusions somewhat overstate and generalize the results and their contrast with previous literature. These need to be more nuanced and more carefully placed within the existing knowledge base.

Specific comments

1) The overall results statements (and conclusions) on retention based on effective Henry's constant values are too broad and overstate the results. The statements should be less sweeping and be more nuanced. Reference to specific lines and discussion is provided below.

L390-391. "retention of compounds with an H*<10^3 is close to 0, i.e. *the entire amount* of the compound dissolved in water is released to the gas phase during riming." And L443 "is negligible" overstate the results seen in the graph. The graph shows retention of about 20% for H* of 10^3, and about 10% for H* of 100. This results statement (and conclusions) should be corrected to not overstate the results, such as by replacing "the entire amount" with "most".

L395. "For compounds with H values above 10^8, a retention of 1 is expected and the compound *remains completely* in the ice phase during freezing". Due to the variability of values found in this study (and expected from theory), this also is too broad a statement. Saying something like "retention of about 1 is expected with most of the compound remaining in the ice phase" would be more appropriate.

Additionally, this threshold value is consistent with that suggested by Stuart and Jacobson (2004) for dry growth riming, who stated "A better parameterization for species with high effective Henry's law constants (the cut-off being somewhere in the range of $10^6$ and $10^{10}$ M/atm) would be to assume complete retention." Providing a comparison to thresholds in the existing literature would be helpful.

L26-27. "retention is negligible for molecules with H* below 10^3, while unity retention can be expected for compounds with H* above 10^8". Again this should be less definitive.

2) The conclusion of no difference in dependencies between the dry and wet growth conditions (L427 "this study shows that there appears to be no difference between dry and wet growth conditions ...") should be more clearly limited by the conditions considered. This is particularly needed regarding the implied disagreement with the conclusions of Michael and Stuart (2009) (L429 "This is in contrast to the modelling study of Michael and Stuart ...).
In the current study, wet growth of graupel with supercooled drop/air surface temperatures (-0.8 to -2.2ºC) and no water shedding was considered whereas Michael and Stuart (2009) considered wet growth of hail, liquid air surface at 0ºC, and shedding of water. The authors do acknowledge there is a difference between the studies' conditions (earlier, in the introduction), but then use language here in the conclusions sections that suggests a generally applicable conclusion that is too broad (no difference in dry and wet growth conditions), and also imply a disagreement in conclusions between the studies. Due to the difference in conditions, there is no clear contrast between the studies' results. The results here are consistent with those from an earlier study (Stuart and Jacobson, 2004), which considered more similar conditions (supercooled drop surface temperatures). It remains experimentally untested which factors affect retention for conditions with little to no surface supercooling, liquid water remaining at the surface, and water shedding. I also don't think it can yet be unequivocally stated that Henry's law constant is the most important under all conditions. It would be more helpful to put these results in the context of the continuum of studies under different conditions, for which the results here refine and extend the understanding to a larger variety of compounds and conditions using rigorous experimental measurements.

Additionally, the substantial unexplained variation in retention in the results here (even for the high H* compounds), are consistent with the potential importance of small variations of freezing conditions on retention, also supporting the existing literature. This should be discussed.

L113. The term 'instantaneous' is not really appropriate. There is expected to be a finite (albeit very short) freezing time, allowing escape of some chemicals (those with lower effective Henry's constants). It would be more appropriate to say 'quick'.

3) There are problems with the description (and potentially the use) of the desorption correction that need to be clarified (or corrected).
P9. Section 2.4
 a. It is confusing to not provide the calculation of the desorption correction coefficient here. I suggest moving the content defining the desorption correction coefficient (including current Equation 2) to here.
 b. The experiment to determine the correction for desorption during flight was apparently only performed on the lowest H* compound. One would expect the amount of

desorption/adsorption during flight to be dependent on H*. Was the desorption correction adjusted for more soluble/less volatile compounds? If so, how was it adjusted? If not, a justification for why it is not expected to matter much is needed.

c. How did the droplet flight time differ between the desorption correction experiments and the retention experiments? If they are substantially different, the correction coefficient would not be valid to correct retention, so this should be provided/discussed.

L195. How does the residence time (2s) of drops in the experiment to determine the desorption correction coefficient compare with that of drops in the retention experiments? (Additionally, what was the time frame of the retention experiment and was it at steady state?)

P9. Two different approaches appear to be used to calculate the retention coefficient (R). In the second approach (Equation 3), the standard ratio (Equation 1) is adjusted by dividing by the desorption coefficient (D) for "compounds with lower retention coefficient" to correct for the possibility of absorption of excess amounts from air during flight or freezing. If absorption occurs, D will be greater than 1 and R will be adjusted down from the uncorrected value. This correction makes sense, but it is not clear why Equation 3 cannot be applied to all compounds, rather than deciding to apply a different method to different compounds. If absorption is small, the adjustment still works, and if desorption occurs during flight instead, R will be adjusted up, which is also appropriate. So why not use Equation 3 for all compounds? Additionally, what was the cutoff for when to use it and when not and why? Further, the statement that the correction was used for "compounds with lower retention coefficient" suggests that retention is primarily dependent on the compound itself, and not the freezing conditions. That has yet to be established, so it would be better to just clarify the conditions for which it was applied and why, rather than this vague description of the choice being compound dependent. Overall, a comparison and discussion of results for D and R with and without 'correcting' for D should be included. Finally, it is confusing to call D a desorption coefficient, but then use it to adjust for absorption, not desorption (and not use it to correct for desorption at all?).

P9. Equations 1 and 2 use the same variables for different entities, which is very confusing and results in it looking like R will always be 1 (if you assume the same variables mean the same thing and substitute into Equation 3, everything cancels out and you get 1?). However, variables that look the same actually have different definitions in these equations. E.g., $c^{sample}\_{compound}$ means the concentration of the compound in the sampled graupel for equation 1, but it means the concentration of the compound in the sampled droplets for equation 2. Please use variable definitions that are distinct to clarify.

P11. Section 3.1. The section presents the desorption coefficients determined from the experiments with droplet flight. D is less than 1, suggesting desorption occurred during the experiment, which is not surprising because this is the lowest H* compound. However, based on the methods presented in section 2.6, it doesn't seem like these Ds are used in

the retention calculations because only absorption seems to be corrected for with Equation 3. Something doesn't correspond. The methods need to be clarified regarding the correction for desorption and absorption.

4) The discussion of pH dependence could be improved.
Table 4. Theoretically, we would only expect pH dependence for compounds with low enough Kh and with pKas near the range of pH studied. Looking at Table 4, the results seem pretty consistent with this. Only retention of 2-nitrophenol was found to depend on pH; it has the lowest Kh and a pKa that is somewhat close to the range of pHs studied. cis-pinic and cis-pinonic acid have pKas in the range of pHs studied, but have much larger Khs. This is interesting and worth noting.

L362. Clarify that no pH dependence was found within the range studied here (4-5.6).

L72-73. As written, this seems to imply that the dependence of retention on the dimensionless effective Henry's law solubility constant (H*) and the dependence on pH of the droplet are independent. However, for chemicals that dissociate (and for pH that are in the range of the pKs), the H* depends on pH. This is minor, but should be clarified.

5) L258. The linearity of the trend is not clear. There are only 2 series of 3 points each and the trend looks curved. Further, the extrapolation from the experimental range of temperatures to the predicted range of temperatures is substantial. It is not clear that the Ds predicted by this extrapolation are meaningful. A theory-based justification for the assumed curve shape and more data are really needed here. (However, if D is not used for desorption, these results could just be eliminated.)

6) L438. What method was used to determine the distinct Henry's law constants of 4-nitrophenol and 2-nitrophenol in Table 4 if the HENRYWIN software predicts the same values?

Technical corrections

L120. "Michael and Stuart (2009) found in their theoretical study that during wet growth conditions H* is not a dominant factor and low retention values were also found for compounds like HCl". This is a bit misleading. We found H* to be important, with retention increasing as H* increased from 300 to 3x10^6, and then leveling off. We also did not study HCl (or any chemical) directly. Rather we studied impact of specific chemical parameters.

L123. Michael and Stuart (2009) did not find the "supercooling of the liquid surface water" was a major determinant, but rather the ice-liquid interface supercooling (and liquid water content) were most important.

L81-82. The sentence "The organic compounds show a dependence on temperature and ventilation" is unclear. I suggest "Retention of the organics compounds shows ..."

L191. A "l" is missing at the end of "nitropheno".

L347. There is a spelling error in the same word.

L370. "insert version used" should be corrected.

---

## Author Comment (AC1)

The manuscript adds to a body of measurements of retention during riming. A retention coefficient increases from zero to one in the limit that volatilization to the gas phase does not occur during riming. In that limit the material is scavenged from the gas phase, via its incorporation in a graupel particle, opening the possibility for vertical transport and removal.

The measurements evaluated in this contribution can help improve understanding of new particle formation (NPF).

More broadly, the investigation casts a spotlight on parameterizing retention in terms of Henry Law solubility.

The authors should consider my critiques and reply with a revised manuscript.

We thank Jefferson Snider for the supportive review and the valuable and constructive comments/suggestions that helped to improve our manuscript. We have carefully revised the manuscript accordingly. Below you will find our point-by-point responses. Reviewer comments and suggestions are written in black, responses in blue. Changes in the manuscript are marked with "".

L36-37

As scientists we are striving to better understand tropospheric chemistry - the associated roles of aerosol and cloud processes – while aiming to reliably model what is happening. That is clear. However, the Introduction seems overly focused on the upper troposphere and on organic compounds. What is depicted in Figure 1 is also important for latitudes other than tropical and for compounds other than IVOC and SVOC. Sulfur dioxide and sulfuric acid fall into the camp of compounds not mentioned at this point in the Introduction. My recommendation is that you adjust somewhat, so that readers are not left with the impression that the motivating uncertainty is only scenarios in deep convective clouds, or in tropical deep convective clouds, or that the uncertainty only applies to the cloud processing of organic compounds.

We thank Jefferson Snider for drawing our attention to the fact that the motivation focused only on organics. We have adapted the text to provide a broader range of background knowledge.

L31: "The rate of NPF formation is strongly dependent on the concentration of low volatile vapors, the temperature and the number of particles that are present. Low-volatility vapors are for example sulfuric acid, which is formed from the reaction of sulfur dioxide and hydroxyl radicals or via oxidation of dimethyl sulfide, as well as highly oxidized organic compounds (Xiao et al., 2023; Williamson et al., 2019; Andreae et al., 2018; Kerminen et al., 2018; Twohy et al., 2002)."

L32-35

The statement that tropical convection does not produce sinks for small particles (and condensable vapors) needs clarity. If sinks (aerosol surface?) are missing, then this can accelerate NPF. But, if tropical convection also removes gaseous precursor, then NPF is decelerated. In this context, what is known about non-tropical convection?

We have adapted the text to provide more clarity. The reviewer is correct in pointing out that retention is also an important factor in non-tropical convection. We have included this in the introduction by referring to aircraft measurements and the associated determination of the scavenging efficiencies of water-soluble compounds.

L34: "A common explanation for the presence of this high number of small particles at high altitudes is the uplift of condensable vapors with simultaneous removal of existing aerosol particles in deep convective clouds. This removal of larger particles reduces the sinks for small particles and condensable vapors, supporting NPF (Clarke et al., 1998). However, Williamson et al. (2019) showed that tropical convection does not lead to uniquely low particle numbers for larger particles. They then argue that there must be a stronger source of condensable vapors at high altitudes in the marine tropics than in other regions and that most of the models used underestimated available organic matter at high altitudes and predict less NPF in these regions. It is therefore important to investigate the possible transport mechanism of organic precursor components that could lead to NPF at high altitudes (Bardakov et al., 2022).

Among other mechanisms, deep convection plays an important role in the transport of trace substances and aerosols into the upper troposphere. In this region, these substances have a longer atmospheric lifetime, thereby increasing the likelihood of long-range transport. Additionally, they can contribute to NPF (Bardakov et al., 2022; Barth et al., 2007a; Barth et al., 2007b). The fraction that arrives in the upper troposphere is influenced by the liquid phase and mixed-phase scavenging of the substances. Aircraft measurements from the USA in thunderstorm inflow and outflow regions demonstrate that water-soluble trace gases, such as $H_2O_2$, are removed with efficiencies between 79% and 97%, which are also influenced by the process of retention (Bela et al., 2018; Barth et al., 2016)."

L53 – 54

It's not clear what is implied by "autoxidation."

Thank you for this comment. We have modified the sentence to the manuscript to hopefully clarify what is implied by autoxidation.

L65: "Highly oxygenated organic molecules (HOMs) exhibit a sufficient low vapor pressure for NPF (Bianchi et al., 2019), however, their formation via autoxidation, a rapid OH-radical–induced oxidation process in the atmosphere, is suppressed at low temperatures (Stolzenburg et al., 2018)"

L85 – 88

In those prior investigations, were drops or droplets collected from regions that were _not _ supercooled? If that was the case, then this statement is not obviously true.

We thank Jefferson Snider for the clarification. It is not clear from the measured samples whether they were supercooled droplets. We have therefore softened our conclusion.

L99: "Measurements of rain, hail and cloud water have already shown that they contain a-pinene oxidation products and nitrophenols (Spolnik et al., 2020; Desyaterik et al., 2013; Ganranoo et al., 2010). It is therefore likely that these compounds are also present in the supercooled droplets within mixed phase zones of clouds."

L103

Is the "simulated graupel" here the same at the "captively floated" target discussed later?

The two terms are used synonymously. The text has been adapted to make this clearer.

L200: "To produce the simulated graupel, a silicon mold was filled with ultra-pure water and frozen. The graupel were "captively floated" to avoid the loss of graupel and any contamination on contact with the wind tunnel walls."

**2.2 Growth Regimes**

During wet growth, broadly speaking, the sample is at ~ 0 oC, droplets are collected, some of that material adds to the mass of the sample, and some is shed. Your observation is that T > -3 oC (this is an ambient temperature threshold, correct?) make for "no freezing." Could this be because the simulated graupel (and the bar) are thermally coupled to a warmer apparatus?

We would like to thank for pointing out this possibility of thermal coupling. The -3°C is the ambient temperature. The Teflon-coated bars are attached to the wind tunnel wall, so it cannot be completely ruled out that the effect may be also due to thermal coupling with warmer surfaces. However, we assume that the effect is more likely to be due to the surface. Teflon is hydrophobic, so that at warmer temperatures the supercooled droplets may not freeze on the hydrophobic surface upon collision.

A comment: Saying that the layer is freezing "very slowly" is confounding an already difficult concept. I will argue that, during wet growth, the freezing rate of an element of input liquid is impossible to calculate. In contrast, during dry growth, freezing rate can be calculated because shape, mass, and boundary conditions are constrained. Rates are fast (the impacted droplet is small, and the temperature gradient is reasonably large) and the characteristic time is small (<< 1 s).

It is correct that "very slowly" is confounding. We have adapted the sentence in the manuscript to provide more clarity.

L129: "During wet growth, the freezing rate of an element of liquid input is rather low in comparison to dry growth conditions."

I like how you have tied with the theoretical work of Michael and Stuart (2009) and brought in your observations of impinged droplets forming larger surface elements. What is the evidence that there is no shedding?

The statement is based on observations and not on a measurement, so we cannot exclude with 100% certainty that shedding has taken place. However, we were unable to observe any visible shedding of droplets. Furthermore, at an inflow velocity of only 3 m/s, which is low compared to the fall velocity of hail in the atmosphere, we do not expect any shedding to occur. We have adapted the sentence to make the statement more specific.

L138: "However, unlike the study of Michael and Stuart (2009), it was not observed that the droplets shed off during these experiments."

L160 –

You present _normalized_ number and _normalized_ mass distributions. Why can't this method be used to quantify LWC?

In principle, it is possible to use the number or the mass distribution to calculate the LWC, but the method would be less accurate. The sample volume is relatively small and therefore the uncertainty is larger. For this reason, we decided to use the method presented in the manuscript, which is more precise.

L174 –

Can you reference a thesis, dissertation, or publication where the distance between the rime collector (s) and the sprayer is documented? If not, please specify that distance.

We would like to thank the reviewer for this comment. The distance is now specified.

L190: "The distance between the sprayer and the experimental section is approx. 3 m (Jost, 2012)."

L180 – 182

This needs better clarity.  The apparatus captured droplets on an impaction substrate where they froze to form rime. Subsequently you melted the sample and measured the concentration of analyte in the liquid.  Please revise for clarity.

We have changed the description a bit and hope that it is now easier to follow.

L190: "In the experimental section, the supercooled droplets collided with three different surfaces, which were used as rime ice collectors and froze on them. (…) The ice samples were collected after each experimental run and stored at -25 °C until they were melted for the chemical analysis."

L191 –

There is a typo in this sentence. The same typo is on L348.

We would like to thank the reviewer for pointing out the typing errors, which have now been corrected.

L200 –

Why were the melted samples filtered?  Is it possible that analyte (or IC) was lost or gained during this step?

We would like to thank Jefferson Snider for pointing out the possible impairment The samples are filtered to remove possible particles, such as dust, and thus protect the HPLC system. We have investigated the influence of filtration, and no significant difference was found.

L208: "For analysis, the ice samples were melted and filtered through polyamide (PA) membranes (pore size: 0.20 µm; Altmann Analytik) to remove potential particles, but without affecting the concentration of the analytes."

L212 –

What is "a. u."?  This occurs twice in this Section.

We thank the reviewer for noticing the missing explanation. a. u. stands for arbitrary unit. The software of the mass spectrometer does not display the nitrogen flow in an actual unit, but only as a numerical value in a. u.. We have added the explanation to the manuscript.

L220: "Sheath gas and auxiliary gas flow was 40 and 20 a. u.(arbitrary unit) respectively."

L234 –

The sentence has "nebulizer" but Figure 2, and the equations, have "sprayer."  These are the same thing, I believe. "Nebulizer" and "nebulization" also appear later in the manuscript. Please use consistent terminology.

Nebulizer and sprayer are used as synonyms. We have carefully gone through the manuscript again to ensure consistent naming.

L248 –

It is important to know if the other nitrophenol isomer, the one with the larger KH, was tested for desportion.

What about other KH < 105 M atm-1 compounds?  For example, was the desportion of pinanediol tested? These KH < 105 M atm-1 compounds partition mostly into the gas phase, assuming phase equilibrium.

We thank the reviewer for this statement. Only the desorption for 2-nitrophenol was determined, as the method with the liquid nitrogen finger was not suitable for this substance. As the other substances including pinanediol all have a retention of around 1, we assumed that desorption has no significant influence on the results. In addition, it was tested for pinic acid that the two measurement methods (using the LN finger or the sprayer blank, with the assumption D = 1) do not differ significantly for substances with high retention coefficients, indicating that desorption has no significant influence for this kind of substances. Therefore, we decided to use the LN finger because it is more accurate (the sample distance is assumed to be too short to allow for a noticeable influence of desorption for compounds with high H*), and we do not need to measure the desorption correction coefficients for all substances.

L250: "Equation 1 and 2 yield identical results for compounds with a desorption correction coefficient of approximately 1. This is illustrated using pinic acid as a representative example in the Supplement (Figure S3)."

L255 –

Longer exposure time in the wind tunnel, with two sprayers compared to four sprayers, is consistent with the discussion of nitrogen volumetric flow rates.  With four sprayers, the nitrogen flow rate is larger, and so, the exposure time is smaller. Based on that, I'm expecting longer exposure time with two sprayers, more liquid-to-gas desportion with two sprayers, and a smaller "desportion correction coefficient" with two sprayers. Larger "desportion correction coefficient", with two sprayers, is contrary to Figure 4.  Because the fit lines are converging toward smaller temperature, this may not matter.  But it's important to get the interpretation of Figure 4 correct.

We want to thank Jefferson Snider for the thorough reading and questioning of the results. The droplets produced when using four sprayers have a longer residence time in the tunnel as a lower nitrogen flow per spray nozzle is used. Even though the total flow rate is higher (24 L/min compared to 20 L/min with two sprayers), this flow rate is divided between four sprayers so that the nitrogen flow rate per sprayer is lower and the exposure time of the droplets is therefore longer. We have included additional measured points in the figure to give a more accurate representation of the data. We now used all the measurements for one linear fit because the slopes and y-axis intercepts of the different nitrogen fluxes are not significantly different, and this gives us better statistics. We have revised the section on desorption based on a comment from Amy L. Stuart. We have decided to include the section on desorption in the supplement rather than the manuscript (see Supplement: desorption correction procedure).

L286 –

A period is missing.

We would like to thank the reviewer for pointing out the missing period, which have now been corrected.

L283 -

How is your result, for pinanediol, an exception to that reported in Jost et al. (2017)? Chemically, the pinanediol (1, 2) is structurally different from the formaldehyde diol.

Pinanediol is an exception, as the data of Jost et al. (2017) indicate that substances with a Henry's law constant comparable to that of pinanediol have a temperature dependence and a lower retention coefficient, which was not the case in our measurements.

Table 4 and its footnotes –

The footnotes, and associated citations, are obscuring this aspect of the presentation. Why can't the US EPA citation be "US EPA, 2012"? Related to this, please see my comment below about the two Henry constants you present (Table 4) for the two structural isomers.

We thank the reviewer for this comment. We cite the US EPA in this way because this is the citation style recommended by the provider.

L436 –

The 2 nitrophenol isomer may form an intramolecular hydrogen bond, possibly weakening its binding to solvent (water). This same rationale might also explain why the Henry constants are so markedly different. There is some discussion of this in the chemical literature. Please provide some insight, and reference citations, in your revision.

We thank the reviewer for this remark. We added a paragraph about this topic to our manuscript.

L448: "The different arrangement allows for the formation of an intramolecular hydrogen bond between the OH and the nitro group in 2-nitrophenol. This can result in the non-dissociated form being stabilized, which may explain why 4-nitrophenol exhibits greater solubility than 2-nitrophenol. This could be due to the fact that 4-nitrophenol undergoes easier solvation and displays the capacity to form intermolecular hydrogen bonds. This property may also be responsible for the observed differences in Henry's law constants and retention (Achard et al., 1996; Schwarzenbach et al., 1988). In contrast to the bond method used in this study, the group method of the HENRYWIN™ software predicts the same Henry's law constant for both isomers."

L438 (and Table 4) –

It is not clear how you arrived at the different Henry constants for the two structural isomers. Did these values come from HENRYWIN? Here (L438) you state that the two isomers are predicted to have the same KH. Also (L365), you state that "The calculated values of H* are listed in Table 4. Since there are no measured Henry's law constants nor reaction enthalpies for some of the more complex organic compounds, these were predicted using the HENRYWIN software which provides the values for 298 K." This is confusing.

Two different prediction methods can be used with the HENRYWIN software. The group method and the bond method. The data in Table 4 and in the figure were all obtained using the bond method. In L438 it is pointed out that it is important to have reliable predictions or measurements for *H\**. For the two nitrophenols, the same value would be obtained using the group method, which does not seem to make physical sense. The text has been adapted so that it is hopefully no longer confusing.

L367: "Since there are no measured Henry's law constants nor reaction enthalpies for some of the more complex organic compounds, these were predicted using the bond method of the HENRYWIN™ software which provides the values for 298 K."

L452: "In contrast to the bond method used in this study, the group method of the HENRYWIN™ software predicts the same Henry's law constant for both isomers. This clearly shows the importance of reliable prediction or measurement of H* and the importance of chemical structure."

---

## Author Comment (AC2)

**Borchers et al. "Retention of a-pinene oxidation products and nitro-aromatic compounds during riming"**

General comments

This is an interesting and important experimental study of retention during riming that supports, refines, and expands the existing knowledge on the phenomenon. It further contributes to understanding and parameterization of the fate of a variety of compounds in freezing clouds, with potential implications for understanding of particle formation and other chemistry of the troposphere. The experimental approach is clever and carefully includes measurement of conditions and properties that have been hypothesized to affect retention, but for which more experimental data are needed. However, some of the calculation methods are unclear and need to be more rigorously explained (or applied), particularly regarding the use of the desorption/absorption correction and its extrapolation. Additionally, the conclusions somewhat overstate and generalize the results and their contrast with previous literature. These need to be more nuanced and more carefully placed within the existing knowledge base.

We thank Amy L. Stuart for her thorough and supportive review of our work. We have carefully revised the manuscript accordingly. Below you will find our point-by-point responses. Reviewer comments and suggestions are written in black, responses in blue. Changes in the manuscript are marked with "".

Specific comments

1) The overall results statements (and conclusions) on retention based on effective Henry's constant values are too broad and overstate the results. The statements should be less sweeping and be more nuanced. Reference to specific lines and discussion is provided below.

L390-391. "retention of compounds with an H*<10^3 is close to 0, i.e. *the entire amount* of the compound dissolved in water is released to the gas phase during riming." And L443 "is negligible" overstate the results seen in the graph. The graph shows retention of about 20% for H* of 10^3, and about 10% for H* of 100. This results statement (and conclusions) should be corrected to not overstate the results, such as by replacing "the entire amount" with "most" .

We would like to thank Amy L. Stuart for pointing out that the statements are too broad. We have carefully reviewed the manuscript and adapted the text.

L393: "Our results show that the retention of compounds with an H* below $10^3$ is close to 0, i.e. most of the compound dissolved in water is released into the gas phase during riming"

L395. "For compounds with H values above 10^8, a retention of 1 is expected and the compound *remains completely* in the ice phase during freezing". Due to the variability of values found in this study (and expected from theory), this also is too broad a statement. Saying something like "retention of about 1 is expected with most of the compound remaining in the ice phase" would be more appropriate.

The statement has been adapted to make it less broad.

L397: "For $H^* > 10^3$ the compounds are present in significant amounts. For compounds with an $H^*$ value above $10^8$, a retention of 1 is expected with most of the compound remaining in the ice phase during freezing."

Additionally, this threshold value is consistent with that suggested by Stuart and Jacobson (2004) for dry growth riming, who stated "A better parameterization for species with high effective Henry's law

constants (the cut-off being somewhere in the range of $10^6$ and $10^{10}$ M/atm) would be to assume complete retention." Providing a comparison to thresholds in the existing literature would be helpful.

We thank for pointing out that the results are consistent with the literature and have included this information in the manuscript.

L399: "This finding is in accordance with the literature. Stuart and Jacobson (2004) suggest a threshold value between $10^6$ and $10^{10}$ M/atm for dry growth riming, which is in the same order of magnitude as the values presented here."

L26-27. "retention is negligible for molecules with H* below 10^3, while unity retention can be expected for compounds with H* above 10^8". Again this should be less definitive.

We have changed the text to make it less definitive.

L25: "Our results reveal that this correlation can also be applied to more complex organic molecules, and that retention under these conditions is not a significant factor for molecules with $H*$ below $10^3$, while retention close to 1 can be expected for compounds with $H*$ above $10^8$."

2) The conclusion of no difference in dependencies between the dry and wet growth conditions (L427 "this study shows that there appears to be no difference between dry and wet growth conditions …") should be more clearly limited by the conditions considered. This is particularly needed regarding the implied disagreement with the conclusions of Michael and Stuart (2009) (L429 "This is in contrast to the modelling study of Michael and Stuart …).

In the current study, wet growth of graupel with supercooled drop/air surface temperatures (-0.8 to -2.2ºC) and no water shedding was considered whereas Michael and Stuart (2009) considered wet growth of hail, liquid air surface at 0ºC, and shedding of water. The authors do acknowledge there is a difference between the studies' conditions (earlier, in the introduction), but then use language here in the conclusions sections that suggests a generally applicable conclusion that is too broad (no difference in dry and wet growth conditions), and also imply a disagreement in conclusions between the studies. Due to the difference in conditions, there is no clear contrast between the studies' results. The results here are consistent with those from an earlier study (Stuart and Jacobson, 2004), which considered more similar conditions (supercooled drop surface temperatures). It remains experimentally untested which factors affect retention for conditions with little to no surface supercooling, liquid water remaining at the surface, and water shedding. I also don't think it can yet be unequivocally stated that Henry's law constant is the most important under all conditions. It would be more helpful to put these results in the context of the continuum of studies under different conditions, for which the results here refine and extend the understanding to a larger variety of compounds and conditions using rigorous experimental measurements.

Thank you for clarifying the differences between the studies. We have updated the text and now present the differences between the studies in the conclusion.

L434: "This study shows that there appears to be no difference between dry and wet growth conditions for compounds with a high effective Henry's law constant and that $H*$ can also be used to estimate retention coefficients for wet growth conditions, at least for graupel and the ambient conditions used in this study (wet growth conditions: ambient temperature: −3 and −5 °C; LWC: 2.2 g m$^{-3}$; surface temperature: −0.8 and −2.2 °C; no shedding of water). This is in contrast to a modelling study by Michael and Stuart (2009), which indicates a lower influence of $H*$ and low retention coefficients even for compounds with high $H*$ under wet growth conditions for hailstones. However, it should be noted that in this study, hail was investigated at a surface temperature of 0°C and a shedding of water. It is

therefore possible that the differences in the experimental conditions are responsible for the discrepancies in the observed outcomes."

Additionally, the substantial unexplained variation in retention in the results here (even for the high H* compounds), are consistent with the potential importance of small variations of freezing conditions on retention, also supporting the existing literature. This should be discussed.

We agree that we did not adequately explain the variation in results. However, the variation in our measurements is in the same range as in other experimental studies. We have added a section to the manuscript where this is discussed.

L268: "The scatter of the values is comparable to that observed in other studies on retention. Theoretical and experimental studies have demonstrated that the retention of compounds with a low H* is dependent on the freezing conditions (Jost et al., 2017; v. Blohn et al., 2013; v. Blohn et al., 2011; Stuart and Jacobson, 2004, 2003). The results shown here indicate that slight differences in freezing conditions across the experiments may exert an influence on the retention of compounds, despite their high H* values."

L113. The term 'instantaneous' is not really appropriate. There is expected to be a finite (albeit very short) freezing time, allowing escape of some chemicals (those with lower effective Henry's constants). It would be more appropriate to say 'quick'.

We have adapted the text and replaced instantaneously with quickly.

L125: "Under dry growth conditions the surface temperature of the rime collector remains well below 0 °C and all the accreted cloud water freezes within some milliseconds on the rime collector, preserving a close to spherical shape."

3) There are problems with the description (and potentially the use) of the desorption correction that need to be clarified (or corrected).

P9. Section 2.4

a. It is confusing to not provide the calculation of the desorption correction coefficient here. I suggest moving the content defining the desorption correction coefficient (including current Equation 2) to here.

Amy L. Stuart is correct in noting that it is confusing to not provide the calculation at this point. We have changed the text to make it easier to follow. However, due to a following comment about the flight time (see c), we have decided to provide the information on desorption in the supplement. (section: Desorption correction procedure).

b. The experiment to determine the correction for desorption during flight was apparently only performed on the lowest H* compound. One would expect the amount of desorption/adsorption during flight to be dependent on H*. Was the desorption correction adjusted for more soluble/less volatile compounds? If so, how was it adjusted? If not, a justification for why it is not expected to matter much is needed.

We thank the reviewer for this statement. Only the desorption for 2-nitrophenol was determined, as the method with the liquid nitrogen finger was not suitable for this substance. The higher gas phase concentration resulting from desorption can lead to adsorption on the liquid nitrogen finger, which would falsify the retention measurement. As the other substances all have a retention of around 1, we assumed that desorption has no significant influence on the results. In addition, it was tested for pinic acid that the two measurement methods (using the LN finger or the sprayer blank, with the assumption

D = 1) do not differ significantly for substances with high retention coefficients, indicating that desorption has no significant influence for this kind of substances. Therefore, we decided to use the LN finger because it is more accurate (the sample distance is assumed to be too short to allow for a noticeable influence of desorption for compounds with high H*), and we do not need to measure the desorption correction coefficients for all substances.

L250: "Equation 1 and 2 yield identical results for compounds with a desorption correction coefficient of approximately 1. This is illustrated using pinic acid as a representative example in the Supplement (Figure S3)."

c. How did the droplet flight time differ between the desorption correction experiments and the retention experiments? If they are substantially different, the correction coefficient would not be valid to correct retention, so this should be provided/discussed.

The reviewer is correct in noting that the time difference between the sampling point for the desorption measurements and the retention measurements is important. The desorption measurements were performed in the lower horizontal part of the tunnel and not where the retention measurements were carried out. Our original assumption—that the droplets are in a quasi-steady state between the desorption measurement and the retention measurement—turns out to be incorrect. Consequently, we have implemented a necessary correction, as detailed in the supplement (section: Desorption correction procedure). As this is beyond the scope of the paper, we will now only present the desorption measurements in the supplementary materials.

L195. How does the residence time (2s) of drops in the experiment to determine the desorption correction coefficient compare with that of drops in the retention experiments? (Additionally, what was the time frame of the retention experiment and was it at steady state?)

We have now measured the previously estimated residence time of the droplets. This is 1 second up to the point where the desorption measurement was performed and 4 seconds up to the region where retention measurements are performed. During this time interval, the droplets are not in a steady state. We have corrected the values for 2-nitrophenol as described in the supplement (Desorption correction procedure) and updated the figures and equations accordingly.

P9. Two different approaches appear to be used to calculate the retention coefficient (R). In the second approach (Equation 3), the standard ratio (Equation 1) is adjusted by dividing by the desorption coefficient (D) for "compounds with lower retention coefficient" to correct for the possibility of absorption of excess amounts from air during flight or freezing. If absorption occurs, D will be greater than 1 and R will be adjusted down from the uncorrected value. This correction makes sense, but it is not clear why Equation 3 cannot be applied to all compounds, rather than deciding to apply a different method to different compounds. If absorption is small, the adjustment still works, and if desorption occurs during flight instead, R will be adjusted up, which is also appropriate. So why not use Equation 3 for all compounds? Additionally, what was the cutoff for when to use it and when not and why? Further, the statement that the correction was used for "compounds with lower retention coefficient" suggests that retention is primarily dependent on the compound itself, and not the freezing conditions. That has yet to be established, so it would be better to just clarify the conditions for which it was applied and why, rather than this vague description of the choice being compound dependent. Overall, a comparison and discussion of results for D and R with and without 'correcting' for D should be included. Finally, it is confusing to call D a desorption coefficient, but then use it to adjust for absorption, not desorption (and not use it to correct for desorption at all?).

Thanks for the comment. Equation 3 (now Eq. 2) corrects for desorption only. The calculations do not include a correction for absorption. The investigated compounds are not present in the wind tunnel air

and could not be detected in blank measurements using only water without additives for droplet formation. Therefore, we can assume that no absorption takes place. However, in the case of highly volatile compounds such as 2-nitrophenol (used in this study) or $SO_2$ (v. Blohn et al. 2013; https://doi.org/10.1007/s10874-013-9261-x) the increased desorption can lead to a higher gas phase concentration and thus to adsorption of the substance on the liquid nitrogen finger. Adsorption on ice (graupel or bars) is unlikely as the surface temperature on these is much higher. Adsorption on the LN finger causes a discrepancy between the measured concentration and the concentration of droplets immediately prior to freezing. The LN finger method (Equation 1) is no longer applicable. The concentration of the solution at the sprayer is used to calculate the retention for compounds with strong desorption. However, since the distance between the sprayer and the experimental region is now several meters, desorption, which causes a loss of substance even before freezing, must be taken into account. The desorption correction coefficient is used for this purpose. However, this correction is not used for all substances, as this would require the desorption to be determined for all substances, which would be very time consuming. This is not necessary for compounds with low desorption because the gas phase concentration is very low and adsorption on the LN finger is therefore unlikely. The method using the LN finger also accounts for the possibility of desorption of the substances. Since the distance to the graupel and bar is very small, the concentration of the droplets can be determined just before the riming. Thus, this method offers good reliability without the need to determine the exact desorption correction factor applicable to each individual compound.

The text has been adapted in the hope that it is now easier to follow.

L237: "The calculation is different for compounds with a lower effective Henry's law constant (below $10^4$), which leads to higher desorption. These compounds are transferred to the gas phase in larger amounts before and during the freezing process, resulting in a higher gas phase concentration in the tunnel. Therefore, it cannot be ruled out that the measured concentrations of the LN finger tube samples could be influenced by additional adsorption out of gas phase components. This would no longer provide a suitable correction for determining the retention coefficient.

As the LN finger sample is not available, the sprayer sample (the solution from which the droplets are produced) is used instead. As the distance between the two sampling points (sprayer and graupel/bar) is large, it is necessary to determine the amount of compound that is transferred to the gas phase before freezing. The desorption correction coefficient $D$ (see Supplement (Eq S3)) will be utilized. A detailed description of the desorption correction coefficient can be found in the Supplement.

P9. Equations 1 and 2 use the same variables for different entities, which is very confusing and results in it looking like R will always be 1 (if you assume the same variables mean the same thing and substitute into Equation 3, everything cancels out and you get 1?). However, variables that look the same actually have different definitions in these equations. E.g., c^sample_compound means the concentration of the compound in the sampled graupel for equation 1, but it means the concentration of the compound in the sampled droplets for equation 2. Please use variable definitions that are distinct to clarify.

The reviewer is correct in stating that using the same variables for different purposes is confusing. The text that is now included in the supplement has been modified to provide a clearer explanation.

Supplement L49: "The numerator describes the ratio of the compound remaining in the sample (droplets sampled in a vial) to a reference sample, which is in this case the sprayer solution, which is the solution immediately before droplet formation, takes place ($c_{compound}^{vial} / c_{compound}^{sprayer}$). The

denominator describes the same ratio but for the IS ($c_{\text{IS}}^{\text{vial}}$ / $c_{\text{IS}}^{\text{sprayer}}$): $D_{\text{compound}} = \frac{c_{\text{compound}}^{\text{vial}} / c_{\text{compound}}^{\text{sprayer}}}{c_{\text{IS}}^{\text{vial}} / c_{\text{IS}}^{\text{sprayer}}}$"

P11. Section 3.1. The section presents the desorption coefficients determined from the experiments with droplet flight. D is less than 1, suggesting desorption occurred during the experiment, which is not surprising because this is the lowest H* compound. However, based on the methods presented in section 2.6, it doesn't seem like these Ds are used in the retention calculations because only absorption seems to be corrected for with Equation 3. Something doesn't correspond. The methods need to be clarified regarding the correction for desorption and absorption.

Thank you for your comment. We believe there may be a misunderstanding. The corrections determined here are utilized to correct the retention of 2-nitrophenol for the desorption that occurs prior to freezing. A more detailed explanation has already been provided in an earlier comment.

4) The discussion of pH dependence could be improved.

Table 4. Theoretically, we would only expect pH dependence for compounds with low enough Kh and with pKas near the range of pH studied. Looking at Table 4, the results seem pretty consistent with this. Only retention of 2-nitrophenol was found to depend on pH; it has the lowest Kh and a pKa that is somewhat close to the range of pHs studied. Cispinic and cis-pinonic acid have pKas in the range of pHs studied, but have much larger Khs. This is interesting and worth noting.

Thank you for this comment. We have adapted the text to draw attention to this point.

L362: "The fact that a pH dependence was only found for 2-nitrophenol is consistent with the literature. A pH dependence is only to be expected for compounds with low H*. Additionally, a dependence is expected primarily in a pH range near the pKa value, as the impact on the ratio between dissociated and non-dissociated molecules is most pronounced in this range (Reijenga et al., 2013; Stuart and Jacobson, 2003). The pKa values of pinic acid and pinonic acid are the most similar to the investigated pH values. However, the H* values seem to be too high to detect an influence of pH on the measurements."

L362. Clarify that no pH dependence was found within the range studied here (4-5.6).

The text has been adapted accordingly.

L358: "Since most of the compounds analyzed in this study did not show any pH dependence within the pH range of 4 and 5.6, a pH value of 4 was used for the calculation of *H\** for these compounds."

L72-73. As written, this seems to imply that the dependence of retention on the dimensionless effective Henry's law solubility constant (H*) and the dependence on pH of the droplet are independent. However, for chemicals that dissociate (and for pH that are in the range of the pKs), the H* depends on pH. This is minor, but should be clarified.

Thanks for pointing this out. We've updated the text to show that H* also depends on physical parameters.

L84: "The retention of a compound is influenced by its chemical properties, such as the dimensionless effective Henry's law solubility constant *H\**, as well as physical parameters such as temperature, droplet size, liquid water content in the cloud, ventilation, and potentially the pH of the droplet, which also influences *H\**."

5) L258. The linearity of the trend is not clear. There are only 2 series of 3 points each and the trend looks curved. Further, the extrapolation from the experimental range of temperatures to the predicted

range of temperatures is substantial. It is not clear that the Ds predicted by this extrapolation are meaningful. A theory-based justification for the assumed curve shape and more data are really needed here. (However, if D is not used for desorption, these results could just be eliminated.)

The reviewer is correct in noting that the trend looks curved. We have included additional measured points in the figure to give a more accurate representation of the data (Figure S5). We now used all the measurements for one linear fit because the slopes and y-axis intercepts of the different nitrogen fluxes are not significantly different, and this gives us better statistics. There is a statistically significant linear trend in temperature. Previous studies in the wind tunnel in Mainz on desorption measurements also show a linear trend that extends to the temperature range of the retention measurements, so we have assumed that this applies to the compound on which we are working. The text has been adapted accordingly and can be found in the supplement.

Supplement L54: "Figure S5 shows the desorption coefficient as a function of the temperature as measured in the lower part of the wind tunnel. Obvious from Fig. S5 is a strong dependence of the desorption coefficient of 2-nitrophenol on temperature, indicating less desorption at lower temperatures. This is expected since the mass transfer rate of a dissolved compound to the environment decreases for decreasing temperatures. The reason for this is that the mass transfer depends, among others on the temperature-dependent parameters like gas- and aqueous-phase diffusivities and the effective Henry's law constant. The observed variations in the values are greater than those expected due to analytical measurement error. These variations are likely attributable to fluctuations in the conditions within the wind tunnel, which were expected to be also present during the retention measurements.

[Figure]

Figure S5: Desorption coefficient for 2-nitrophenol. The errors correspond to the error of the analytical measurement (y-axis) or the standard deviation of the temperature measurement during the collection time of each sample (x-axis). The dashed line represents a linear regression.

A statistically significant linear trend is visible in the measurement series. A linear regression with the equation $D_{\mathrm{compound}} = a_{\mathrm{D}} \cdot x \cdot [°C]^{-1} + b_{\mathrm{D}}$ was performed to extrapolate the desorption to the

temperatures present during the experiments. This leads to the following equation, S2, for the fit function.

$$D_{\text{compound}} = (-0.020 \pm 0.002) \cdot x \cdot [°C]^{-1} + (0.606 \pm 0.015) \quad\quad\quad \text{(S2)}$$

Extrapolating the results to the experimental conditions (e.g. -10 °C and -4 °C) yields $D_{\text{compound}} = 0.81$ and $D_{\text{compound}} = 0.67$ for typical dry and wet growth conditions respectively."

6) L438. What method was used to determine the distinct Henry's law constants of 4-nitrophenol and 2-nitrophenol in Table 4 if the HENRYWIN software predicts the same values?

We used the bond method because the group method would give us the same value for both substances. We updated the text to clarify this.

L452: "In contrast to the bond method used in this study, the group method of the HENRYWIN™ software predicts the same Henry's law constant."

Technical corrections

L120. "Michael and Stuart (2009) found in their theoretical study that during wet growth conditions H* is not a dominant factor and low retention values were also found for compounds like HCl". This is a bit misleading. We found H* to be important, with retention increasing as H* increased from 300 to 3x10^6, and then leveling off. We also did not study HCl (or any chemical) directly. Rather we studied impact of specific chemical parameters.

The text has been adapted so that it is hopefully no longer misleading.

L134: "For example, Michael and Stuart (2009) found in their theoretical study that during wet growth conditions H* is an important but not a dominant factor. They observed that retention increased with increasing H* (from 300 to $3 \cdot 10^6$) and then leveled off with further increasing H*, resulting in low retention values for compounds with high H* such as HCl."

L123. Michael and Stuart (2009) did not find the "supercooling of the liquid surface water" was a major determinant, but rather the ice-liquid interface supercooling (and liquid water content) were most important.

The text has been corrected.

L137: "Here other factors like the ice-liquid interface supercooling and the liquid water content are major determinators for the extent of retention."

L81-82. The sentence "The organic compounds show a dependence on temperature and ventilation" is unclear. I suggest "Retention of the organics compounds shows …"

The sentence has been changed.

L94: "Retention of the organic compounds show a dependence on temperature and ventilation."

L191. A "l" is missing at the end of "nitropheno".

L347. There is a spelling error in the same word.

L370. "insert version used" should be corrected.

Thank you for pointing this out. The spelling mistakes and the incorrectly placed "insert used version" have been corrected.